# Mechanism of bidirectional thermotaxis in *Escherichia coli*

Anja Paulick[1], Vladimir Jakovljevic[2], SiMing Zhang[3], Michael Erickstad[4], Alex Groisman[4], Yigal Meir[5], William S Ryu[3], Ned S Wingreen[6], Victor Sourjik[1,2]*

[1]Max Planck Institute for Terrestrial Microbiology and LOEWE Research Center for Synthetic Microbiology, Marburg, Germany; [2]Zentrum für Molekulare Biologie der Universität Heidelberg, Heidelberg, Germany; [3] Department of Physics and Donnelly Centre, University of Toronto, Toronto, Canada; [4]Departments of Physics, University of California, San Diego, United States; [5]Department of Physics, Ben Gurion University of the Negev, Beer Sheva, Israel; [6]Department of Molecular Biology, Princeton University, Princeton, United States

**Abstract** In bacteria various tactic responses are mediated by the same cellular pathway, but sensing of physical stimuli remains poorly understood. Here, we combine an in-vivo analysis of the pathway activity with a microfluidic taxis assay and mathematical modeling to investigate the thermotactic response of *Escherichia coli*. We show that in the absence of chemical attractants *E. coli* exhibits a steady thermophilic response, the magnitude of which decreases at higher temperatures. Adaptation of wild-type cells to high levels of chemoattractants sensed by only one of the major chemoreceptors leads to inversion of the thermotactic response at intermediate temperatures and bidirectional cell accumulation in a thermal gradient. A mathematical model can explain this behavior based on the saturation-dependent kinetics of adaptive receptor methylation. Lastly, we find that the preferred accumulation temperature corresponds to optimal growth in the presence of the chemoattractant serine, pointing to a physiological relevance of the observed thermotactic behavior.

DOI: https://doi.org/10.7554/eLife.26607.001

*For correspondence:
victor.sourjik@synmikro.mpi-marburg.mpg.de

Competing interests: The authors declare that no competing interests exist.

## Introduction

For many organisms temperature is one of the crucial environmental factors that determine growth and fitness. Thus, it is not surprising that organisms developed sophisticated systems for sensing and responding to temperature (*Sengupta and Garrity, 2013*). Indeed, the capability to detect and follow environmental temperature gradients – thermotaxis – is inherent to many organisms, from animals to bacteria. Although in eukaryotes temperature is usually sensed by specific thermal sensors, behavioral controls by temperature and chemical stimuli are tightly intertwined in the well-studied examples of *Caenorhabditis elegans* (*Kimata et al., 2012*) and *Drosophila melanogaster* (*Montell, 2013*; *Ni et al., 2013*). Such integration of behavioral responses is even more pronounced in bacteria. In *Escherichia coli* the thermotactic and chemotactic responses are mediated by the same pathway (*Maeda and Imae, 1979*; *Maeda et al., 1976*). The tactic behavior of *E. coli* generally relies on the control of flagellar motors by a signaling pathway that decreases the rate of 'tumbles' (reorientations) upon an increase in the levels of attractants or upon a decrease in the levels of repellents. As a result cells make longer runs in the preferred direction, resulting in a net propagation up attractant gradients (*Brown and Berg, 1974*; *Macnab and Koshland, 1972*). Stimulus recognition relies on signaling complexes that consist of transmembrane receptors, the scaffold protein CheW, and the histidine kinase CheA (*Gegner et al., 1992*; *Hazelbauer et al., 2008*; *Sourjik, 2004*). Conformational changes that are induced by binding of chemical ligands in the periplasm are transmitted to

**eLife digest** Many bacteria can move towards or away from chemicals, heat and other stimuli in their environment. The ability of bacteria to move in response to nutrients and other chemicals, known as chemotaxis, is the best understood of these phenomena. Bacteria generally swim in a fairly random way and frequently change direction. During chemotaxis, however, the bacteria sense changes in the concentrations of a chemical in their surroundings and this biases the direction in which they swim so that they spend more time swimming towards or away from the source of the chemical. The bacteria have various receptor proteins that can detect different chemicals. For example, the Tar and Tsr receptors can recognize chemicals called aspartate and serine, respectively, which are – amongst other things – nutrients that are used to build proteins.

Tar and Tsr are also involved in the response to temperature, referred to as thermotaxis. At low temperatures, a bacterium *Escherichia coli* will move towards sources of heat. Yet when the bacteria detect both serine and aspartate they may reverse the response and move towards colder areas instead. However, it was not clear why the bacteria do this, and what roles Tar and Tsr play in this response.

Paulick et al. have now combined approaches that directly visualise signalling inside living bacteria and that track the movements of individual bacterial cellswith mathematical modelling to investigate thermotaxis in *E. coli*. The experiments show that the bacteria's behaviour could be explained by interplay between the responses mediated by Tar and Tsr. In the absence of both serine and aspartate, both receptors stimulate heat-seeking responses, causing the bacteria to move towards hotter areas. When only aspartate is present, Tsr continues to stimulate the heat-seeking response, but the aspartate causes Tar to switch to promoting a cold-seeking response instead. This leads to the bacteria accumulating in areas of intermediate temperature. In the presence of serine only, the bacteria behave in a similar way because the receptors swap roles so that Tsr stimulates the cold-seeking response, while Tar promotes the heat-seeking one.

The intermediate temperature at which the bacteria accumulate in response to serine is also around the optimal temperature for *E.coli* growth in presence of this chemical, suggesting that thermotaxis might play an important role in allowing bacteria to survive and grow in many different environments, including in the human body. Thus, understanding how chemotaxis and thermotaxis are regulated may lead to new ways to control how bacteria behave in patients and natural environments.

DOI: https://doi.org/10.7554/eLife.26607.002

the cytoplasmic part of the receptor and control the activity of CheA, which is inhibited by attractants and stimulated by repellents (*Falke and Hazelbauer, 2001*; *Hazelbauer et al., 2008*). CheA-mediated phosphorylation of the response regulator CheY stimulates its binding to the flagellar motor and induces tumbling, whereas dephosphorylation of CheY by the phosphatase CheZ promotes smooth swimming. Adaptation to persistent stimuli in the chemotaxis system is mediated by the methyltransferase CheR and the methylesterase CheB, which adjust the level of receptor methylation and thereby receptor activity dependent on the background stimulation (*Goy et al., 1977*; *Springer et al., 1979*). The system functions as an integral negative feedback circuit, whereby CheR preferentially methylates inactive receptors, thus increasing their activity, whereas CheB preferentially demethylates active receptors (*Barkai and Leibler, 1997*; *Shapiro et al., 1995*; *Terwilliger et al., 1986*; *Yi et al, 2000*).

The most abundant receptors in *E. coli* are Tsr and Tar, which respectively sense the amino acids serine and aspartate but can also detect other stimuli (*Adler et al., 1973*; *Greer-Phillips et al., 2003*; *Kondoh et al., 1979*; *Maeda et al., 1976*; *Mesibov and Adler, 1972*; *Slonczewski et al., 1982*; *Springer et al., 1979*). In the cell, all chemoreceptors form large mixed clusters in the inner membrane, where cooperative interactions between multiple receptors serve to amplify chemotactic stimuli (*Ames et al., 2002*; *Briegel et al., 2012*; *Sourjik and Berg, 2004*; *Studdert and Parkinson, 2004*; *Zhang et al., 2007*). The coupling of neighboring receptors within clusters also allows integration of signals perceived by different types of receptors, so that the net response of a cooperative signaling unit is determined by the net of the free-energy changes due to stimulation of individual

receptors (*Keymer et al., 2006*; *Mello and Tu, 2005*; *Neumann et al., 2010*). Notably, although the activities of different receptors are tightly coupled, adaptation to stimuli results in preferential methylation of the stimulus-specific receptor (*Lan et al., 2011*).

Chemotaxis is typically assumed to enable bacteria to find conditions that are optimal for growth, and correlation between chemotactic and metabolic preferences has indeed been observed for *E. coli* (*Yang et al., 2015*). Consistent with that, *E. coli*'s response to chemical ligands is usually unidirectional, that is cells either follow gradients only upwards (attractant) or only downwards (repellent). In contrast, gradients of other stimuli that are sensed by the chemotaxis pathway, such as pH (*Yang and Sourjik, 2012*), osmolarity (*Adler et al., 1988*), or temperature (*Maeda et al., 1976*), might need to be followed bidirectionally in order to find optimal growth conditions. However, while the unidirectional chemotactic response of *E. coli* is comparatively well understood, mechanisms of bidirectional taxis in bacteria remain to be established.

*E. coli* is well known to follow temperature gradients and to react to temporal changes in temperature, similar to the tactic response to chemical effectors (*Maeda et al., 1976*). This tactic response to temperature is mainly mediated by Tar and Tsr (*Imae et al., 1984*; *Lee et al., 1988*; *Maeda and Imae, 1979*; *Mizuno and Imae, 1984*), although minor receptors Trg, Aer, and Tap might also be temperature-sensitive (*Nara et al., 1991*; *Nishiyama et al., 2010*). The thermotactic behavior of *E. coli* is thus primarily determined by the interplay of the responses mediated by Tar and Tsr (*Yoney and Salman, 2015*). At low temperatures, *E. coli* is normally thermophilic (heat-seeking), with an increase in temperature causing an attractant-like response (*Maeda et al., 1976*). However, the thermotactic response of *E. coli* to higher temperatures, 36°C to 42°C, remained ambiguous, with different studies showing either loss of response or its inversion to cryophilic (*Maeda et al., 1976*; *Paster and Ryu, 2008*; *Yoney and Salman, 2015*). Adaptation to a combination of attractants sensed by Tsr (serine) and Tar (aspartate or its non-metabolizable analogue α-methyl-DL-aspartate, MeAsp) can clearly invert the thermotactic response (*Imae et al., 1984*; *Salman and Libchaber, 2007*), which is likely due to increased methylation of adapted receptors since amino acid replacement of methylation sites also altered thermotactic responses of Tar and Tsr (*Nara et al., 1996*; *Nishiyama et al., 1999a*; *Nishiyama et al., 1999b*; *Oleksiuk et al., 2011*). Several studies showed that Tar functions as a warm sensor in low methylation states but as a cold sensor in high methylation states, whereas Tsr was suggested to similarly function as a warm sensor in low methylation states but to lose its temperature sensitivity in high methylation states (*Imae et al., 1984*; *Salman and Libchaber, 2007*; *Yoney and Salman, 2015*). In contrast, other work showed that both receptors mediate thermophilic response in the low-modification states (zero or one methyl groups per receptor monomer) and cryophilic response in the high-modification states (*Oleksiuk et al., 2011*).

Despite this clear evidence that the thermotactic behavior of *E. coli* depends on both the ambient temperature and chemotactic stimuli, the interpretation of *E. coli* behavior in gradients of temperature is complicated by the fact that, besides the signaling pathway, temperature also affects cell swimming, respiration, and metabolism (*Demir et al., 2011*; *Maeda et al., 1976*; *Salman et al., 2006*). Hence, the key questions whether *E. coli* is capable of accumulation at a specific intermediate temperature solely by means of thermotaxis, and the mechanism of such accumulation, remain unanswered. Importantly, a standard mathematical model of the chemotactic network, developed by Barkai and Leibler (*Barkai and Leibler, 1997*), and generalized to allow for coupled teams of receptors (*Keymer et al., 2006*; *Mello and Tu, 2005*; *Sourjik and Berg, 2004*), cannot explain the accumulation temperature. Although a model that can account for the bidirectional taxis (termed 'precision sensing') was proposed (*Jiang et al., 2009*), this model critically relies on an *ad hoc* assumption about the temperature dependence of the pathway activity that was not experimentally verified.

In this study we investigated the thermotactic response of *E. coli* at the level of the intracellular pathway activity, thus independently of any direct temperature effects on motility. Additionally, we analyzed cell behavior using microfluidic devices that were designed to minimize the time cells spend in temperature gradients, thus reducing secondary effects on cell physiology. The results obtained with both assays were consistent, demonstrating that thermotactic behavior of *E. coli* can indeed be explained solely by specific receptor-mediated responses. We showed that in the absence of chemical attractants the response of *E. coli* is always thermophilic although it is weakened with increasing ambient temperature. By contrast, inversion of the pathway response from thermophilic to cryophilic at intermediate temperature (*inversion* temperature) and bidirectional cell accumulation towards intermediate temperature (*accumulation* temperature) were observed when the cells were

adapted to ligands sensed by either Tar or Tsr, but not when both ligands were present at similar levels. Our results are consistent with the hypothesis that the mechanism of bidirectional thermotaxis relies on the interplay between Tar and Tsr receptors in different methylation states, and we employed a mathematical analysis to elucidate the details of the underlying mechanism. Finally, we demonstrate that the preferred accumulation temperature observed in the presence of serine roughly corresponds to the optimal growth temperature, suggesting that the thermotactic behavior of *E. coli* could indeed be explained by growth-rate optimization.

## Results

### Thermotactic response of *E. coli* depends on ambient temperature

To investigate the thermotactic response of *E. coli* at the level of the pathway activity, we utilized an in-vivo assay based on Förster (fluorescence) resonance energy transfer (FRET) (*Neumann et al., 2012*; *Sourjik and Berg, 2002a*; *Sourjik et al., 2007*) (*Figure 1—figure supplement 1A*). The FRET assay relies on phosphorylation-dependent interactions between CheY and CheZ, which are fused to yellow and cyan fluorescent proteins, respectively. The formation of the CheY-YFP/CheZ-CFP complex, which is proportional to the kinase activity of CheA, leads to an increase in the ratio of YFP to CFP fluorescence due to energy transfer from CFP to YFP. In our FRET experiments, Δ*cheY-cheZ* cells expressing the CheY-YFP/CheZ-CFP FRET pair were exposed to rapid stepwise changes in temperature while under a constant flow of buffer (*Figure 1—figure supplement 1A*). Note that to facilitate the measurements we used a strain deleted for *flgM*, the negative regulator of flagellar and chemotaxis gene expression, that elevates in proper proportion the levels of chemotaxis proteins and thereby enhances the chemotactic response (*Kollmann et al., 2005*; *Steuer et al., 2011*). Similar to the response observed upon stimulation with a chemical attractant in the same setup (*Figure 1—figure supplement 1C*), the FRET response to an increase in temperature revealed a rapid transient decrease of the YFP/CFP ratio, reflecting a decrease in the kinase activity, that is a thermophilic response (*Figure 1A* and *Figure 1—figure supplement 1D*). This transient response was specific because it was not observed in a Δ*cheA* strain (*Figure 1—figure supplement 1E*). A decrease in temperature resulted in an opposite response, that is, a transient increase in FRET, similar to the removal of attractant (*Figure 1A* and *Figure 1—figure supplement 1C*) or the addition of repellent. The time course of subsequent adaptation in the presence of persistent stimulation was also similar for thermal and chemical stimuli, indicating that the adaptation to temperature similarly relies on the CheR/CheB receptor methylation system. This was directly confirmed by measuring the methylation profile of Tsr and Tar, both of which shifted towards higher-methylated states at higher temperature (*Figure 1—figure supplement 2*). The only major noticeable difference between thermal and chemical stimulation in the FRET experiments was an increase in the basal YFP/CFP ratio at higher temperature, which was also seen in the Δ*cheA* strain and is caused by temperature dependence of YFP and CFP fluorescence (*Figure 1—figure supplement 1E*) (*Kumar and Sourjik, 2012*; *Oleksiuk et al., 2011*).

We subsequently used FRET to measure the response to 3°C incremental steps of temperature in the range from 21°C to 42°C (*Figure 1B*). We observed that the thermophilic response of wild-type cells persisted at a similar level up to 30°C, but decreased rapidly at higher ambient temperatures. Nevertheless, even at the highest tested temperature (jump from 39°C to 42°C), the response remained weakly thermophilic. Hence, for cells that were adapted in the buffer (in the absence of chemoattractants) we did not observe any inversion of the pathway response to temperature.

These results were generally consistent with the behavioral response of motile cells in a thermal gradient established across a microfluidic channel (*Figure 1C* and *Figure 1—figure supplement 1B* and *Figure 1—figure supplement 3*). The design of the experiment was such that cells in the sample volume experienced only a brief exposure to a temperature gradient, thus minimizing secondary effects of temperature that might have complicated the interpretation of previous studies. Here, when adapted in the buffer, cells accumulated towards the warmer side of the gradient in the channel, consistent with thermophilic behavior (*Figure 1C*). We quantified this behavior using a thermal migration coefficient (TMC) (*Figure 1C Inset*, see Materials and methods for details). In line with the FRET-based pathway-activity analysis, this thermophilic behavior weakened in the range of higher temperatures but never inverted. Thus, the results of both FRET and microfluidics assays clearly

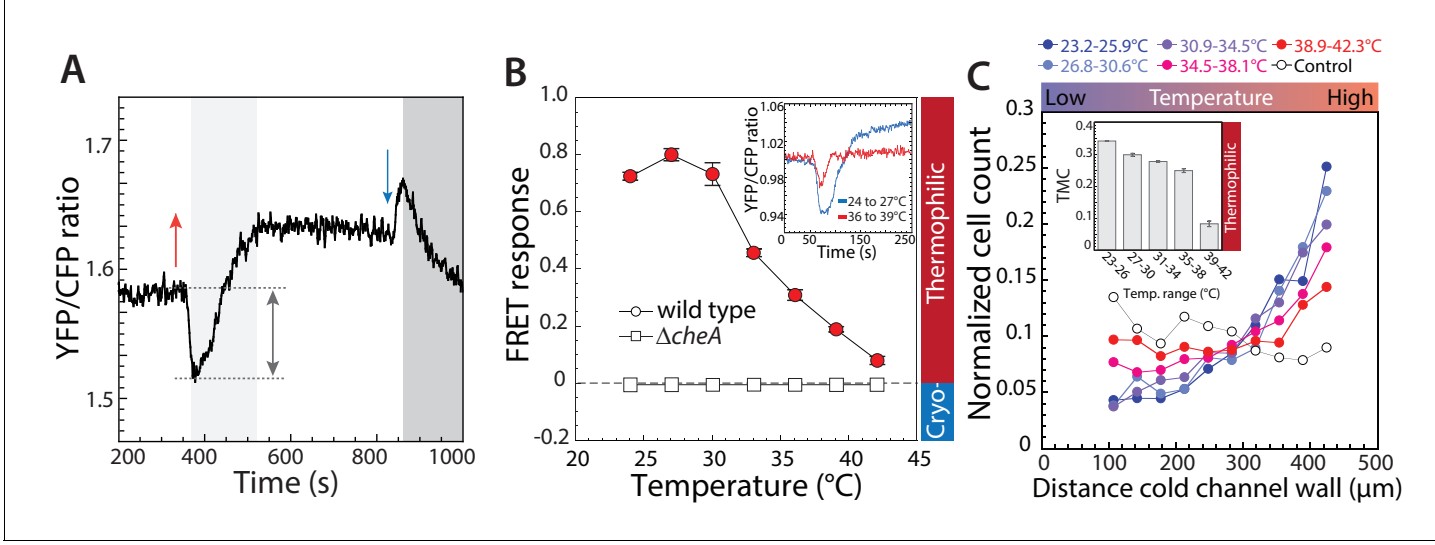

**Figure 1.** Thermotactic response of wild-type *E.coli* in absence of chemical ligands. (A,B) The intracellular kinase activity of CheA was measured using an in-vivo CheY phosphorylation assay based on fluorescence resonance energy transfer (FRET) between CheY-YFP and CheZ-CFP (see Materials and methods). (A) A typical FRET measurement, where the YFP/CFP ratio reflects decrease of CFP fluorescence and increase of YFP fluorescence due to energy transfer within a complex formed by phosphorylated CheY-YFP and CheZ-CFP, with complex formation proportional to the activity of CheA. The initial response of wild-type cells adapted in buffer to a temperature increase of 9°C (red arrow) is an attractant-like response (i.e., decrease in CheA activity), while the subsequent decrease of temperature (blue arrow) elicits a repellent-like response (i.e., increase in CheA activity). CheA activity is subsequently restored to the basal level by CheR and CheB-dependent methylation (light grey area) and demethylation (dark grey area). Note the general increase in the basal YFP/CFP ratio with temperature is due to its differential effect on YFP and CFP fluorescence. (B) The amplitude of the initial response (grey arrow in A) was used to quantify the pathway response of buffer-adapted cells to a 3°C stepwise increment of temperature in wild-type and Δ*cheA* cells as indicated. The response was normalized to the maximal response obtained upon stimulation with saturating concentration (1 mM) of the chemical attractant MeAsp at 21°C. For data points in all figures of this type, thermophilic responses (downregulation of the YFP/CFP ratio) are highlighted in red, cryophilic responses (upregulation of the YFP/CFP ratio) are highlighted in blue, and no significant response is indicated in white. Means of at least three independent experiments and the corresponding standard errors are shown as a function of initial temperature for each step. Inset: An example of the FRET response to an increase of temperature from 24° to 27°C (blue) and from 36° to 39°C (red). Here, for convenience the mean initial YFP/CFP ratio was normalized to one for both measurements. (C) Quantification of the thermotactic response of buffer-adapted cells in a microfluidics device with indicated temperature gradients, with lower temperature at the left channel wall. As a control (white circles) no gradient was applied. Cell counts at different positions of the microfluidic channel were determined and the data were normalized as described in Materials and methods. The inset shows the thermal migration coefficient (TMC) that characterizes drift in the temperature gradient, calculated from three independent experiments (such as that shown in the main panel). Positive values of TMC correspond to thermophilic response, whereas negative values of TMC correspond to cryophilic response.

DOI: https://doi.org/10.7554/eLife.26607.003

The following source data and figure supplements are available for figure 1:

**Source data 1.** Source data for *Figure 1B*.
DOI: https://doi.org/10.7554/eLife.26607.007
**Source data 2.** Source data for *Figure 1C*.
DOI: https://doi.org/10.7554/eLife.26607.008
**Source data 3.** Source data for *Figure 1—figure supplement 2B*.
DOI: https://doi.org/10.7554/eLife.26607.009
**Figure supplement 1.** Thermotaxis assays for *E.coli*.
DOI: https://doi.org/10.7554/eLife.26607.004
**Figure supplement 2.** Temperature dependence of receptor methylation in buffer.
DOI: https://doi.org/10.7554/eLife.26607.005
**Figure supplement 3.** Calibration of the microfluidics device.
DOI: https://doi.org/10.7554/eLife.26607.006

show that in the absence of chemotactic stimuli, *E. coli* has an exclusively thermophilic response that decreases at high ambient temperatures, but it does not actively avoid high temperature.

## Asymmetric chemotactic stimulation leads to accumulation temperature in wild-type cells

We next systematically investigated previously reported inversion of the thermotactic response from thermophilic to cryophilic upon adaptation to high concentrations of serine and aspartate (or the non-metabolizable analogue of aspartate, MeAsp) (*Imae et al., 1984*; *Paster and Ryu, 2008*). Firstly, we measured the pathway response to temperature changes in cells that were stimulated with a combination of MeAsp and serine (*Figure 2A*). These two attractants were kept at a fixed ratio of 10:1, which reflects an approximately tenfold lower chemoattractant efficiency of MeAsp compared to serine (*Neumann et al., 2010*). Indeed, steady pre-stimulation with high levels of both attractants inverted the response to cryophilic over the entire range of temperatures, with the response again becoming weaker at high temperatures. At low concentrations of attractants the response remained thermophilic and no avoidance of high temperature was observed but the response amplitude was reduced compared to buffer-adapted cells. Adaptation to intermediate levels of serine and MeAsp completely abolished the thermotactic response over the entire range of tested temperatures, meaning that the pathway becomes temperature-insensitive when both major chemoreceptors are stimulated at an approximately equal intermediate level.

As a next step, we investigated the effects of adaptation to different levels of either MeAsp or serine alone. In contrast to previous reports (*Imae et al., 1984*; *Salman and Libchaber, 2007*; *Yoney and Salman, 2015*), we observed that adaptation to MeAsp (*Figure 2B*) or serine (*Figure 2C*) had nearly identical effect on the thermotactic response. As in the case of stimulation with a mixture of serine and MeAsp, we observed that adaptation to either attractant weakened the thermophilic response in a dose-dependent manner and could eventually invert it to cryophilic. However, the pattern of this inversion by individual attractants was clearly different. Whereas combined stimulation either inverted or abolished the thermotactic response over the entire temperature range (*Figure 2A*), cells adapted to individual attractants showed cryophilic response at high temperature but retained thermophilic response at low temperatures (*Figure 2B,C*). Such an inversion ('cross-over') temperature, where the FRET response changes from thermophilic to cryophilic, implies that *E. coli* can indeed bidirectionally accumulate towards a preferred temperature using the chemotaxis pathway. However, the inversion and accumulation temperatures may not be exactly identical due to weak direct effects of temperature on *E. coli* motility (*Oleksiuk et al., 2011*).

These conclusions were confirmed by microfluidic experiments, where a combination of serine and MeAsp changed the response to cryophilic (*Figure 2D*) and where, in the presence of individual attractants, the response turned from thermophilic at low temperatures to cryophilic at high temperatures (*Figure 2—figure supplement 1*). This latter inversion of the thermotactic response dependent on the ambient temperature implies that in a thermal gradient and in presence of either one of the major chemoattractants cells should accumulate at some transition temperature (which is expected to be close to the inversion temperature in the FRET assay), being attracted to it from both lower and higher temperatures. Such accumulation was indeed observed in the presence of serine in a gradient that spanned a range of temperatures with both thermophilic and cryophilic responses (*Figure 2E* and *Figure 2—figure supplement 2*). Notably, no cell accumulation was observed in the same thermal gradient in the absence of attractant stimulation (*Figure 2—figure supplement 2A*), meaning that it is not simply due to the wider temperature range used in these experiments.

## Thermosensing properties of Tar and Tsr are similar

Our observation that chemotactic stimulation of Tar or Tsr individually – but not together – creates an accumulation temperature indicates that the effect might be related to the interplay between the two receptors. To test this conclusion, we investigated the thermosensing properties of cells that express only one type of receptor, either Tar (*Figure 3A*) or Tsr (*Figure 3B*) using the FRET assay. Similar to wild-type cells, in the absence of chemoattractants both Tar- and Tsr-only cells exhibited thermophilic responses that decreased with ambient temperature. Adaptation to their respective attractants also decreased and inverted the thermophilic response in a dose-dependent manner, which was again similar for both receptors. Notably, the inversion to cryophilic response occurred over the entire temperature range and no significant cross-over from thermophilic to cryophilic response occurred with the change of temperature at any given concentration of serine or MeAsp.

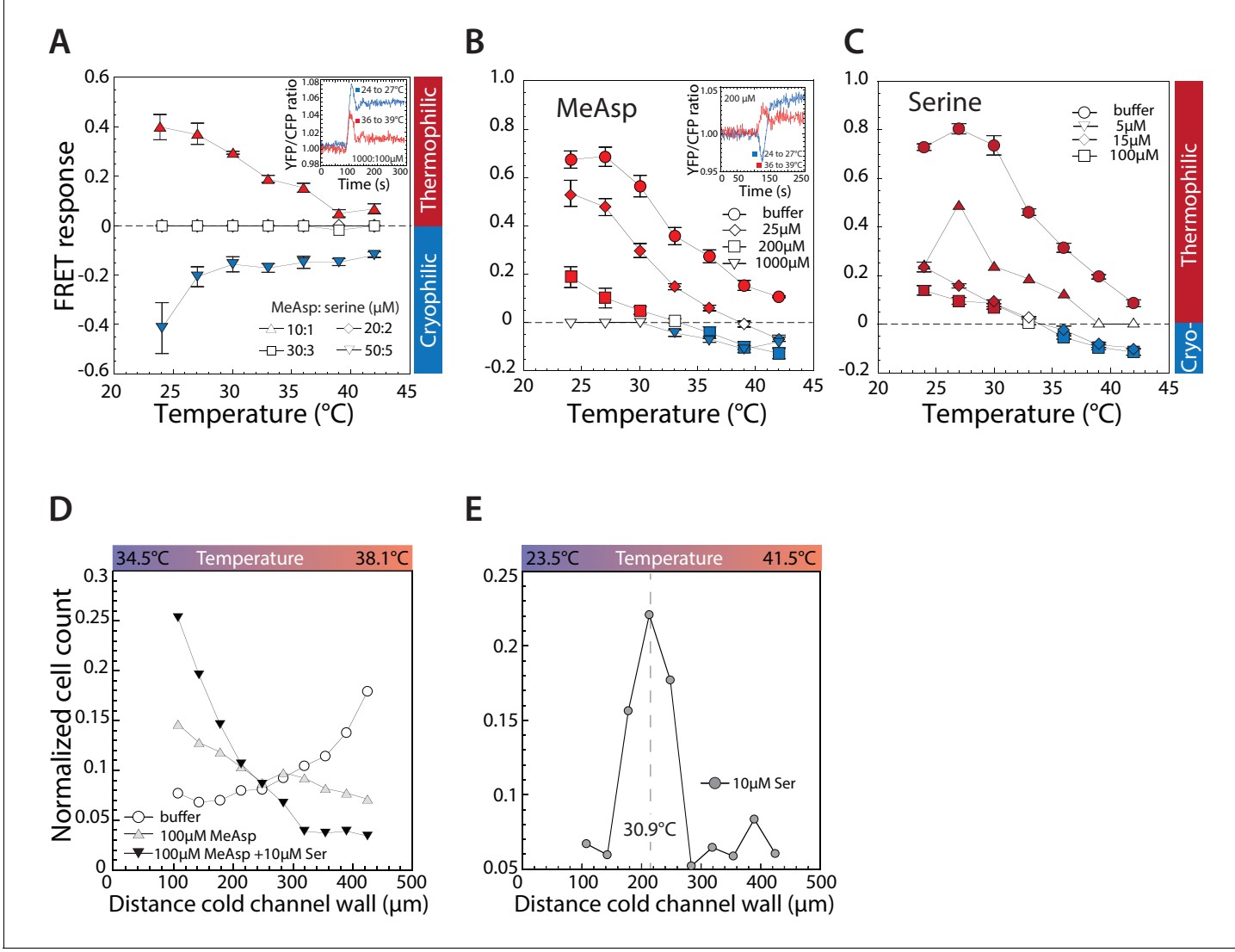

**Figure 2.** Thermotactic response of wild-type *E.coli* in presence of chemical ligands. (A–C) FRET measurements of the pathway response to 3°C steps of temperature after adaptation to a combination of indicated concentrations (in µM) of MeAsp and serine (A); only to MeAsp (B); or only to serine (C). At a 10:1 ratio of MeAsp:serine in (A) both receptors are stimulated approximately equally. Insets: Examples of individual measurements for indicated temperature steps and attractant concentrations. Thermophilic and cryophilic responses are indicated by red and blue symbols, respectively. Absence of response is indicated in white. Data are means of at least three independent experiments and the respective standard errors are displayed. (D,E) Thermotactic responses in a microfluidics device, measured as in *Figure 1C* for wild-type cells adapted to indicated levels of serine and or MeAsp in a 34.5° to 38.1°C temperature gradient (D) or adapted to 10 µM serine in a 23.5° to 41.5°C temperature gradient (E). Dashed line in (E) indicates cells' accumulation point in the gradient.

DOI: https://doi.org/10.7554/eLife.26607.010

The following source data and figure supplements are available for figure 2:

**Source data 1.** Source data for *Figure 2A*.
DOI: https://doi.org/10.7554/eLife.26607.013
**Source data 2.** Source data for *Figure 2B*.
DOI: https://doi.org/10.7554/eLife.26607.014
**Source data 3.** Source data for *Figure 2C*.
DOI: https://doi.org/10.7554/eLife.26607.015
**Source data 4.** Source data for *Figure 2D*.
DOI: https://doi.org/10.7554/eLife.26607.016
**Source data 5.** Source data for *Figure 2E*.
DOI: https://doi.org/10.7554/eLife.26607.017

*Figure 2 continued on next page*

*Figure 2 continued*

**Source data 6.** Source data for *Figure 2—figure supplement 1A*.
DOI: https://doi.org/10.7554/eLife.26607.018
**Source data 7.** Source data for *Figure 2—figure supplement 1B*.
DOI: https://doi.org/10.7554/eLife.26607.019
**Source data 8.** Source data for *Figure 2—figure supplement 1C*.
DOI: https://doi.org/10.7554/eLife.26607.020
**Source data 9.** Source data for *Figure 2—figure supplement 1D*.
DOI: https://doi.org/10.7554/eLife.26607.021
**Source data 10.** Source data for *Figure 2—figure supplement 1E*.
DOI: https://doi.org/10.7554/eLife.26607.022
**Source data 11.** Source data for *Figure 2—figure supplement 2A*.
DOI: https://doi.org/10.7554/eLife.26607.023
**Source data 12.** Source data for *Figure 2—figure supplement 2B*.
DOI: https://doi.org/10.7554/eLife.26607.024
**Source data 13.** Source data for *Figure 2—figure supplement 2C*.
DOI: https://doi.org/10.7554/eLife.26607.025
**Figure supplement 1.** Asymmetric stimulation of receptors leads to thermophilic response at low temperature and cryophilic response at high temperature.
DOI: https://doi.org/10.7554/eLife.26607.011
**Figure supplement 2.** Stimulation of Tsr leads to accumulation in a temperature gradient.
DOI: https://doi.org/10.7554/eLife.26607.012

The responses of the Tar and Tsr-only cells were thus similar to the response of wild-type cells adapted to combinations of serine and MeAsp (*Figure 2A*), but different from adaptation to only one of these chemoattractants (*Figure 2B,C*). These observations confirm that chemotactic stimulation of Tar or Tsr can either inhibit or invert their thermosensing properties. The results also clearly demonstrate that in the presence of only one receptor type there is no temperature-dependent response inversion at any given level of chemotactic stimulation, and thus no accumulation temperature.

The observed dependence of the thermotactic response on ambient temperature and chemotactic stimulation is likely to be explained by the known effects of methylation on receptor thermosensing (*Imae et al., 1984*; *Nara et al., 1996*; *Nishiyama et al., 1997*; *Paster and Ryu, 2008*; *Salman and Libchaber, 2007*). Our results are consistent with a previous observation suggesting that the interplay between methylation and thermosensing is similar for Tar and Tsr (*Oleksiuk et al., 2011*), which both mediate thermophilic response in low-methylation states but cryophilic response in high-methylation states. For the buffer adapted cells, where the level of receptor modification is low (*Endres et al., 2008*), the response of both receptors is thus thermophilic. However, because adaptation to positive (attractant-like) temperature stimuli is mediated by increased methylation of receptors (*Figure 3C*, *Figure 1—figure supplement 2* and *Figure 3—figure supplement 1*), the thermophilic response becomes weaker at higher temperature. In contrast, for cells of single-receptor strains adapted at high concentration of the respective chemoattractant, the methylation level is high and the response is thus cryophilic. Adaptation to this repellent-like response leads to increased demethylation of receptors at higher temperature (*Figure 3C*), thus weakening the cryophilic response. At intermediate levels of chemotactic stimulation that correspond to zero thermotactic response, there are no associated changes in receptor methylation and therefore no switch from thermophilic to cryophilic behavior.

As mentioned above, our results for Tsr are apparently in contrast with some previous studies. Specifically, Yoney and Salman (*Yoney and Salman, 2015*) reported gradual dose-dependent inhibition of the thermophilic response but no response inversion to cryophilic response when cells that express Tsr as the only major receptor were stimulated with glycine, a Tsr-specific attractant. This discrepancy can be explained by the much lower apparent affinity of glycine to Tsr compared to serine (*Yang et al., 2015*). Indeed, in FRET experiments we found that at glycine concentrations of up to 1 mM, the temperature response of the Tsr-only strain is either thermophilic or non-detectable (*Figure 3—figure supplement 2A,B*). The response of the Tsr-only strain became weakly cryophilic

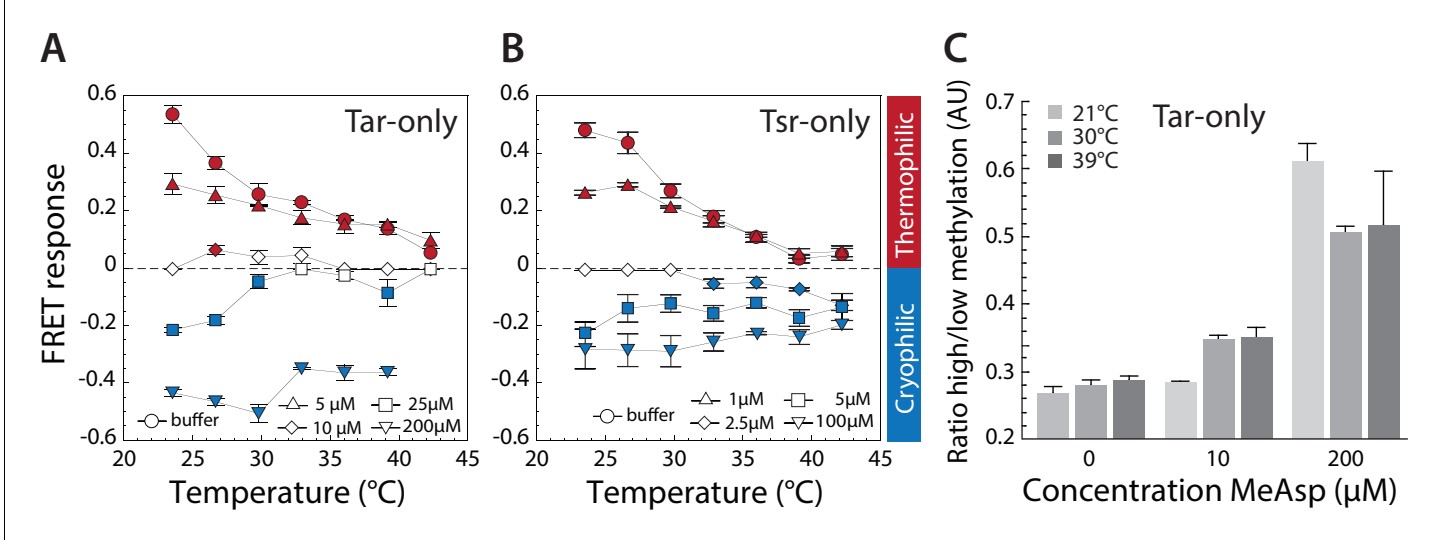

**Figure 3.** Thermotactic response of Tar- and Tsr-only strains. (A,B) FRET measurements of the pathway response in receptorless *E. coli* strains expressing Tar (A) or Tsr (B) as the sole receptor from a plasmid. Cells were adapted to the depicted concentrations of MeAsp (A) or serine (B). Thermophilic, cryophilic or no responses are highlighted in red, blue and white, respectively. Data are means of at least three independent experiments and the standard error is displayed. (C) Methylation levels of Tar in Tar-only cells adapted to buffer or indicated levels of MeAsp at the depicted temperatures. Methylation was determined according to receptor mobility in SDS-PAGE and quantified as the ratio of high to low states of methylation (see *Figure 3—figure supplement 1*). Data points are mean values from three independent experiments, with error bars representing standard errors.

DOI: https://doi.org/10.7554/eLife.26607.026

The following source data and figure supplements are available for figure 3:

**Source data 1.** Source data for *Figure 3A*.
DOI: https://doi.org/10.7554/eLife.26607.032
**Source data 2.** Source data for *Figure 3B*.
DOI: https://doi.org/10.7554/eLife.26607.033
**Source data 3.** Source data for *Figure 3C*.
DOI: https://doi.org/10.7554/eLife.26607.034
**Source data 4.** Source data for *Figure 3—figure supplement 3*.
DOI: https://doi.org/10.7554/eLife.26607.035
**Figure supplement 1.** Methylation profiles of Tar-only cells at varying temperature and MeAsp levels.
DOI: https://doi.org/10.7554/eLife.26607.027
**Figure supplement 2.** Tsr-mediated response in presence of glycine.
DOI: https://doi.org/10.7554/eLife.26607.028
**Figure supplement 3.** Serine is depleted from M9CG medium upon incubation with *E. coli* cells.
DOI: https://doi.org/10.7554/eLife.26607.029
**Figure supplement 4.** Inversion of thermotactic response depends on cell culture density.
DOI: https://doi.org/10.7554/eLife.26607.030
**Figure supplement 5.** Indirectly binding ligands do not invert thermotactic response.
DOI: https://doi.org/10.7554/eLife.26607.031

only when the concentration of glycine was increased to a very high level of 30 mM (*Figure 3—figure supplement 2C*). Moreover, Yoney and Salman reported accumulation of the wild-type *E. coli* cells in a thermal gradient that was established in a complex medium that contained a mixture serine and aspartate (*Yoney and Salman, 2015*). Notably, in these experiments the exposure of bacterial culture to a thermal gradient was preceded by a prolonged incubation, which most likely resulted in substantial depletion of serine from the medium as it is rapidly consumed by *E. coli* (*Yang et al., 2015*). Such depletion could indeed be confirmed (*Figure 3—figure supplement 3* and Supplementary material), suggesting that during the thermotaxis assays, cells were primarily stimulated by only one major chemoattractant, aspartate. The observed behavior is thus consistent with our conclusion that asymmetric stimulation of either Tsr or Tar is required for *E. coli* accumulation at an intermediate temperature.

## Ratio between Tar and Tsr affects response inversion

Our results strongly suggest that an accumulation temperature arises from the interplay between Tar and Tsr when only one type is strongly stimulated by its ligand. This suggests that the level of chemotactic stimulation that leads to the response inversion as well as the accumulation temperature might be affected by the relative expression levels of Tar and Tsr. Because these levels are known to vary with the growth phase of an *E. coli* culture, with Tsr being more abundant during the early to mid-exponential phase (*Kalinin et al., 2010*; *Yang and Sourjik, 2012*), we tested the thermotactic response in cultures grown to different optical densities (*Figure 3—figure supplement 4*). Indeed, we observed that cells adapted to high concentrations of serine showed earlier response inversion (i. e., inverted at lower temperature) when grown to low optical density, and did not invert at all when grown to high optical density. This observation may explain why no inversion of the thermal response upon adaptation to serine was observed in a previous study (*Imae et al., 1984*) hat used *E. coli* culture grown to high density. The opposite dependence on the growth phase was observed for MeAsp-stimulated cells, which inverted at higher temperature when grown to low optical density.

Here we only considered the interplay between Tar and Tsr in defining the overall thermotactic response of wild-type cells. While other receptors might also be temperature sensitive (*Nara et al., 1991*; *Nishiyama et al., 2010*), their low abundance in *E. coli* as compared to the abundances of Tar and Tsr (*Li and Hazelbauer, 2004*) makes it unlikely that they significantly contribute to the thermotactic response of wild-type cells. Indeed, adaptation to the ligands of Trg (glucose, galactose) or Tap (dipeptides) had no noticeable effect on the thermotactic response (*Figure 3—figure supplement 5*).

## Mathematical model of accumulation in a thermal gradient

How can we understand the experimental observation of an accumulation temperature? As mentioned above, the standard model of the chemotactic network cannot explain the observed accumulation of *E. coli* towards a specific temperature when both Tsr and Tar receptors are present with one type stimulated by attractant. To explain this accumulation, we therefore developed a minimal model for the activity of chemotaxis receptors, based on the one detailed in Meir *et al.* (*Meir et al., 2010*) (see *Source code file 1* - Modelling). The aim of the model is to provide insight into the origin of an accumulation temperature, not to quantitatively account for the experimental data which depends on many unknown temperature-dependent parameters. The basic elements of the model are (i) a free-energy model for the probability that a team of chemoreceptors will be active, depending on the methylation level of the receptors and the concentration of ligand and (ii) a kinetic model for the rate of change of the receptor methylation level due to the enzymes CheR and CheB. Importantly, this model incorporates the failure of precise adaptation when the number of available methylation or demethylation sites becomes small (*Meir et al., 2010*), which we believe is essential to explain the accumulation temperature. Some indication of the importance of imprecise adaptation is already apparent for the single-receptor cells (*Figure 3A,B*) – the response in the presence of high ligand concentration is flatter as a function of temperature, consistent with the assumption of slowing down of methylation near saturation. Consistent with our experimental results and with previous work (*Oleksiuk et al., 2011*), we assume for simplicity identical behavior of Tar and Tsr, except for their different ligand specificities.

To most simply illustrate the physical origin of the accumulation temperature for wild-type cells, we chose the basic signaling unit (team) of allosterically interacting receptors to be a trimer, where Tar and Tsr are randomly mixed so that each of the three receptors can be either Tsr or Tar with probability reflecting relative expression level (*Ames et al., 2002*; *Hansen et al., 2010*) (see Materials and methods). As clear from the comparison of the data (*Figure 2B,C* and *Figure 3A*) to the results of the mathematical model, the model captures correctly the complete inversion of the thermal response form thermophilic to cryophilic with methylation level or ligand concentration (*Figure 4A*) for a single type of receptor, as well as cross-over (i.e., temperature-dependent inversion) of the response for mixed receptors (*Figure 4B*). Within the model, these qualitative features can be understood as follows (for a more detailed discussion see Supplementary material): As discussed above, for a single receptor type the thermal response has either one sign or the other over all temperatures, with the sign determined by methylation level (which depends on the attractant concentration). Changes of temperature alone can never change the sign of the thermal response

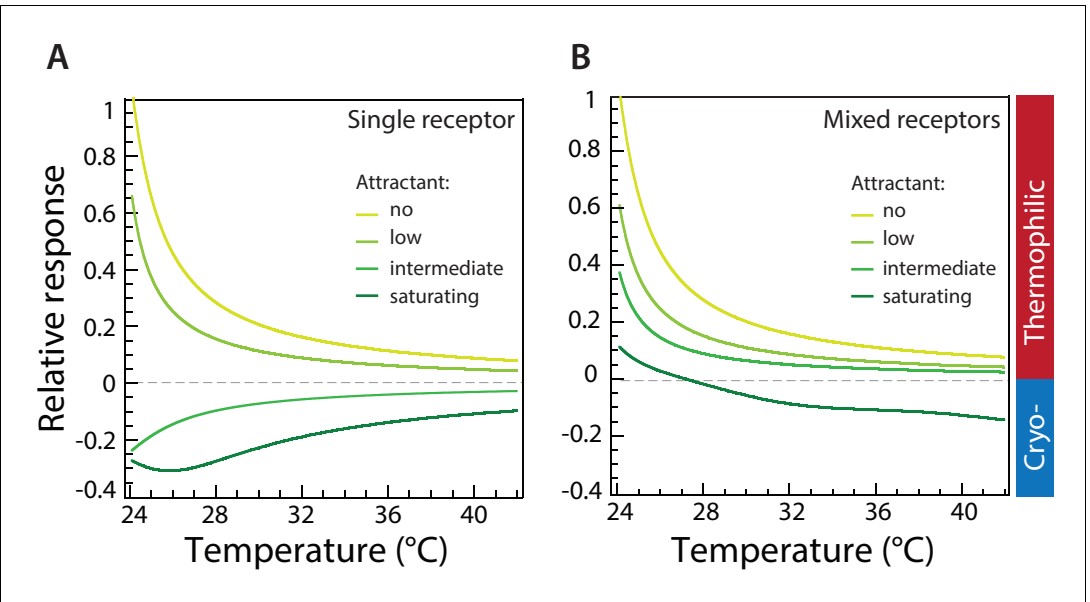

**Figure 4.** Computer simulations of thermotactic response. (**A**) Thermotactic response of the chemosensory complexes was simulated for a single type of receptor (Tar) using a mathematical model described in Materials and methods. Compare to experimental data in *Figure 3A*. The thermotactic response is shown as a function of temperature. Simulations were performed in the absence of attractant (MeAsp) or in the presence of low, intermediate, or saturating concentrations of MeAsp, as indicated by different colors. (**B**) Simulated response for randomly mixed complexes of two receptor types (Tar and Tsr), with only Tar being stimulated by attractant. Compare to experimental data in *Figure 2B*. Simulations were performed as in (**A**) at varying levels of stimulation with MeAsp and assuming random mixing of equal amounts of Tar and Tsr within receptor trimers. The MeAsp concentration determines the values of the free-energy parameter $f_0$ for Tar, which are (0, 0.5, 1.5, 2.5), respectively (see Materials and methods) in (**A**), and (0, 1, 1.5, 15) in (**B**). The value of $f_0$ for Tsr in (**B**) is 0, independent of MeAsp concentration; other parameters are as specified in Materials and methods. Sidebar indicates regions corresponding to thermophilic or cryophilic responses. Note that the response inversion from thermophilic to cryophilic at saturating MeAsp stimulation in (**B**) is due to imperfect adaptation; no inversion is observed in case of perfect adaptation (see *Figure 4—figure supplement 2*).

DOI: https://doi.org/10.7554/eLife.26607.036

The following figure supplements are available for figure 4:

**Figure supplement 1.** Average modeled methylation levels of receptors versus temperature for different concentrations of chemical attractant.

DOI: https://doi.org/10.7554/eLife.26607.037

**Figure supplement 2.** Modeled thermotactic response of mixed-receptor cells assuming perfect adaptation.

DOI: https://doi.org/10.7554/eLife.26607.038

because if the thermal response approaches zero, as necessary for a sign change, so necessarily does the adaptive change in methylation. Since methylation level determines the sign of the thermal response, no change in methylation means no change in the sign of the thermal response, and thus no inversion of the response at a certain temperature (*Figure 4A* and *Figure 4—figure supplement 1A*). This conclusion also holds for a mixture of different receptors provided adaptation is perfect and thermosensing properties of receptors are identical; as the net thermal response approaches zero, so does the net change in methylation level. Even if one type of receptor becomes more methylated, in the standard model this is exactly compensated by the other type becoming less methylated, and as a consequence the response remains either thermophilic or cryophilic over the entire temperature range (*Figure 4—figure supplement 2*). However, an inversion of the response at a certain temperature becomes possible if changes of temperature lead to a net change in receptor methylation. This is exactly what happens in our mathematical model: when receptors of one type are near the saturation level of their methylation, they cannot adapt perfectly, that is their methylation level changes only weakly upon temperature stimulation, and thus cannot compensate for the

response of receptor of the other type, which undergo an opposite and larger change in their methylation level (*Figure 4—figure supplement 1B,C*). The resulting net change in total methylation of the receptor system can readily produce a change in the sign of the overall thermal response, leading to an inversion of the response at a certain temperature. The actual value of the inversion temperature depends on receptor methylation levels and is thus a function of attractant concentrations. Note that our model for the inversion temperature relies on exactly the same slowing of methylation rates near saturation previously introduced to explain the failure of precise adaptation in chemotaxis (*Meir et al., 2010*), and it requires no further *ad-hoc* assumptions about differences between thermosensing properties of Tar and Tsr or asymmetry between thermophilic and cryophilic responses. This contrasts with the model of Jiang et al. (*Jiang et al., 2009*), which attributes an accumulation temperature to temperature-dependent adapted activity, but which does not explain why an inversion temperature only occurs when both Tar and Tsr are present and then in the presence of ligand for only one of the two receptor types.

## Accumulation temperature correlates with optimal growth

What is the physiological significance of the accumulation temperature observed upon stimulation with amino acid attractants? A previous study proposed that a cryophilic response in the presence of amino acids might be a form of quorum sensing behavior, whereby amino acid secretion at high density would cause cell accumulation at lower temperatures thus slowing metabolism (*Salman and Libchaber, 2007*). However, the benefit of such behavior is not obvious and under normal growth conditions *E. coli* does not secrete chemoattractive amino acids – instead these amino acids are the first to be consumed from the medium (*Prüss et al., 1994*; *Selvarasu et al., 2009*; *Yang et al., 2015*).

At the same time, high concentrations of several amino acids, most notably of serine, were shown to have a growth-inhibitory effect (*Amos and Cohen, 1954*; *Neumann et al., 2014*; *Rowley, 1953*; *Yang et al., 2015*). We thus investigated this toxicity of serine for *E. coli* MG1655 in M9 glycerol minimal medium as a function of growth temperature. We observed that the effect of serine on *E. coli* growth was indeed temperature-dependent: whereas at low temperature (24°C) serine is growth-promoting (*Figure 5A*), it becomes inhibitory at high temperature (39°C) (*Figure 5B*). This temperature dependence could be seen both for the growth delay (*Figure 5C*), where the addition of serine led to a prolonged phase of slower growth at higher temperature, and for the maximal growth rate (*Figure 5—figure supplement 1*). By comparison, the effect of aspartate on growth was much weaker. Interestingly, the observed delay in growth in the presence of serine was minimal at 30°C (*Figure 5C*). These results indicate that at least for cells adapted in the presence of serine, the emergence of an accumulation temperature might have a simple physiological meaning, being an adaptive mechanism that enables *E. coli* to optimize its growth in the presence of this amino acid.

## Discussion

Temperature critically affects growth, metabolism, and other biological processes, and many organisms are capable of following gradients of temperature in their environment. In this work, we clarify the mechanism that enables *E. coli*, which is commonly used as a model of bacterial behavior, to accumulate at a preferred ambient temperature using bidirectional thermotaxis.

Our results show that the entire complexity of the thermotactic behavior of *E. coli* can be accounted for by the interplay between the effects of temperature on the activity of the major chemoreceptors Tar and Tsr and the dependence of receptor methylation on both ambient temperature and chemoeffector stimulation: (i) When cells are adapted in the buffer or in the presence of low levels of Tar and Tsr chemoattractants, both major receptors mediate thermophilic responses, leading to an overall thermophilic response. This response becomes weaker at higher temperature, because the increased levels of methylation make both receptors less thermophilic, consistent with previous analyses (*Nishiyama et al., 1999a*; *Nishiyama et al., 1997*). However, the response in this case cannot invert if the temperature is the sole stimulus, since weaker response also leads to smaller increase in methylation, meaning that there is no driving force to increase receptor methylation beyond the level that makes receptors temperature-insensitive. (ii) When Tar and Tsr are both stimulated with intermediate levels of their respective attractants, their intermediate methylation state makes them – and therefore also wild-type cells – temperature insensitive. Since there is no change

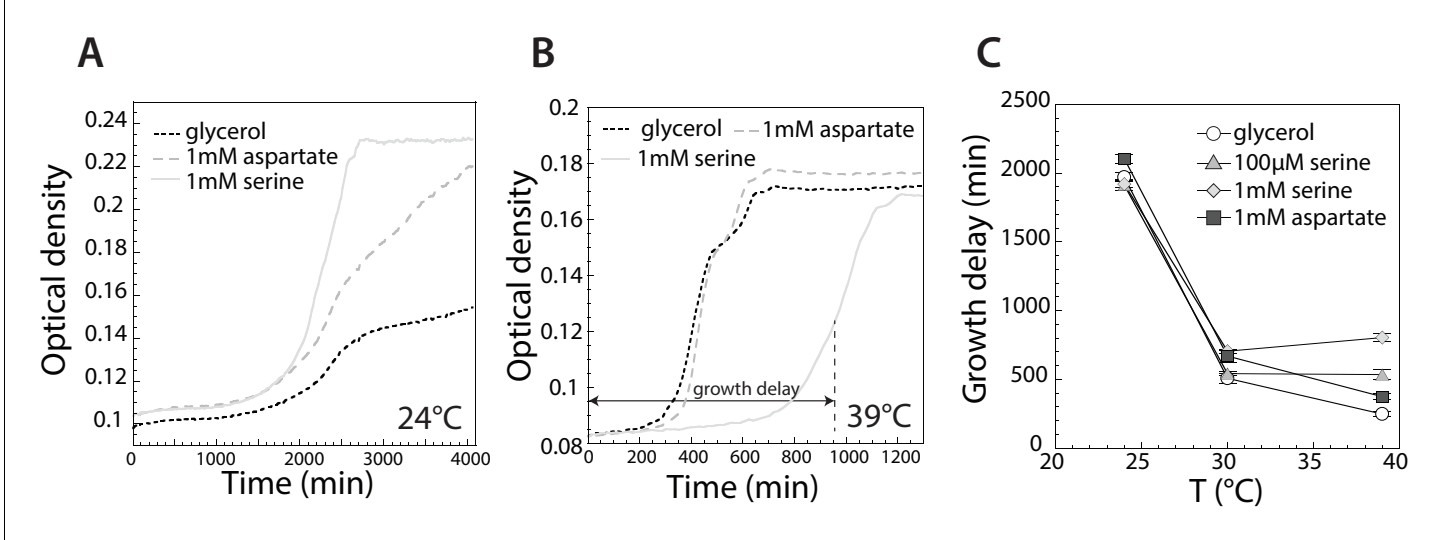

**Figure 5.** Effect of serine and aspartate on cell growth at different temperatures. (A,B) Growth curves of *E. coli* MG1655 cells grown at 24°C (A) and 39°C (B) in M9 minimal medium supplemented with glycerol (0.2%) or with glycerol and 1 mM serine or aspartate, as indicated. Optical density was measured at 600 nm in the plate reader as described in Materials and methods. (C) Time until maximal growth rate achieved (highlighted by the grey arrow in (B)) was used to measure the delay in growth at indicated temperatures and amino acid concentrations. Data are means of at least six independent experiments and the respective standard error is displayed.

DOI: https://doi.org/10.7554/eLife.26607.039

The following source data and figure supplement are available for figure 5:

**Source data 1.** Source data for *Figure 5A*.
DOI: https://doi.org/10.7554/eLife.26607.041
**Source data 2.** Source data for *Figure 5B*.
DOI: https://doi.org/10.7554/eLife.26607.042
**Source data 3.** Source data for *Figure 5C*.
DOI: https://doi.org/10.7554/eLife.26607.043
**Source data 4.** Source data for *Figure 5—figure supplement 1*.
DOI: https://doi.org/10.7554/eLife.26607.044
**Figure supplement 1.** Growth rate dependence on amino acids and temperature.
DOI: https://doi.org/10.7554/eLife.26607.040

in methylation in the absence of a thermotactic response, no inversion from thermophilic to cryophilic response (or *vice versa*) occurs. (iii) Adaptation to high levels of both Tar and Tsr ligands makes both receptors and hence the wild-type response cryophilic. Adaptation to high temperature in the cryophilic regime results in the reduction of methylation and thus weakens the cryophilic response. Such symmetric regulation of the thermal response for Tar and Tsr is in contrast to previous studies that suggested that only Tar can function both as a warm and cold sensor with Tsr acting either as a warm sensor or not responding (*Imae et al., 1984*; *Salman and Libchaber, 2007*). Notably, the cryophilic response observed at high levels of ligand stimulation shows less dependence on ambient temperature than the thermophilic response. (iv) Finally, upon adaptation to either one of Tar or Tsr ligands, only the cognate receptor is preferentially methylated (*Lan et al., 2011*) and thus inverted in its thermosensing. Although in nature *E. coli* is unlikely to be stimulated by a single amino acid, the mechanism proposed here applies as long as the concentrations of serine and aspartate in the medium are strongly different. Such a scenario is consistent with a recent study suggesting that *E. coli* chemotaxis has evolved to primarily follow gradients of individual amino acids rather than of amino acid mixtures (*Yang et al., 2015*).

We further demonstrate that this thermotactic behavior can be accounted for by a mathematical model that describes clusters of mixed receptors as a dynamical and partially decoupled lattice (*Hansen et al., 2008*; *Yang and Sourjik, 2012*). Our analysis suggests that the cross-over from thermophilic to cryophilic response relies on the previously characterized failure of precise adaptation in

bacterial chemotaxis (*Meir et al., 2010*; *Neumann et al., 2014*). This failure occurs at high levels of receptor methylation, which lowers the rate of further methylation. Because of this, positive and negative changes in methylation of the attractant-stimulated Tar and unstimulated Tsr (or *vice versa*) upon a change of temperature are not precisely compensatory, which enables receptor teams to cross the zero point of the thermotactic response. Importantly, our model does not require any *ad-hoc* assumptions of differences between the thermosensing behavior of Tar and Tsr or of the temperature dependence of other pathway parameters, and it can naturally explain the apparent difference between the thermophilic and cryophilic responses as a function of ambient temperature. Consistent with this model of the interplay between Tar and Tsr and with previous observations (*Yoney and Salman, 2015*), we demonstrate that the inversion of the thermotactic response by individual attractants strongly depends on the ratio between these two receptors, which itself is a function of the growth phase of the cell culture (*Kalinin et al., 2010*; *Salman and Libchaber, 2007*; *Yang and Sourjik, 2012*).

This work thus elucidates the mechanism of bacterial thermotaxis and its control by chemical stimuli. Although largely consistent with previous studies on thermotaxis, including a recent work that emphasized the importance of the interplay between Tar and Tsr (*Yoney and Salman, 2015*), surprisingly our results demonstrate that bidirectional thermotaxis requires asymmetric stimulation of either one of the two major receptors, Tar or Tsr. This finding clearly contrasts with previous studies that did not observe inversion of the Tsr response to temperature (*Imae et al., 1984*; *Salman and Libchaber, 2007*; *Yoney and Salman, 2015*). These differences are likely to be explained first by the very low signaling strength of glycine (*Yang et al., 2015*), the Tsr attractant used in two of these studies (*Salman and Libchaber, 2007*; *Yoney and Salman, 2015*), and second by the weaker cryophilic response mediated by Tsr that apparently eluded detection by Mizuno and Imae (*Imae et al., 1984*). It was also proposed that both receptors need to be stimulated with attractants to enable accumulation at a specific temperature (*Yoney and Salman, 2015*). However, this conclusion relied on experiments that were performed upon prolonged incubation of the bacterial suspension in the growth medium containing a mixture of amino acids. Because under these conditions serine is rapidly depleted (*Yang et al., 2015*) (*Figure 3—figure supplement 3*) and glycine is only a weak attractant, these bacteria were likely asymmetrically stimulated by aspartate. This finding can reconcile the previously observed accumulation behavior with the mechanism proposed in our study, further emphasizing the importance of precise control of medium composition in thermotaxis experiments. Secretion of attractants into the medium during long (up to 3.5 hr) pre-stimulus incubation may also explain the inversion of the thermotactic response at high temperature observed by Paster and Ryu (*Paster and Ryu, 2008*).

Finally, our growth experiments also indicate that the observed thermotactic behavior could be at least partly interpreted as a strategy to achieve optimal growth. The optimal growth temperature for *E. coli* is normally assumed to be ~39°C, and we indeed observed that the growth rate in minimal medium with glycerol or glucose as a carbon source steadily increases up to this temperature. Therefore, the thermophilic drift up to this temperature in a thermal gradient would promote cell growth. Nevertheless, this thermophilic drift is gradually slowed down, apparently preventing cell movement to even higher temperatures that would elicit a heat-shock response, although we did not observe active avoidance of high temperature by *E. coli* under these conditions. However, temperature dependence of cell growth was also affected by addition of amino acids. While the addition of serine, and to some extent also of aspartate, was beneficial for growth at 24°C, serine had a strong inhibitory effect on growth at higher temperatures. This inhibitory effect was previously described (*Neumann et al., 2014*), and it might be caused by isoleucine or SAM starvation (*Hama et al., 1991*; *Zhang et al., 2010*; *Zhang and Newman, 2008*). However, the temperature dependence of this inhibition had not yet been demonstrated, and we propose that this dependence might explain the avoidance of high temperature by *E. coli* in the presence of high levels of serine. Interestingly, in our experiments the minimal growth delay in the presence of serine was achieved around 30°C, roughly matching the preferred accumulation temperature under these conditions. Thus, the bidirectional taxis might enable *E. coli* to optimize its growth rate in a thermal gradient.

The mechanism of bidirectional thermotaxis described here for *E. coli* is likely to be relevant not only for closely related species such as *Salmonella* that also possess Tar and Tsr but also for other bacteria and even archaea. Although no other bacterial thermotactic responses have been characterized so far, conservation of receptor sequences across prokaryotic chemotaxis systems makes it likely

that many of them can respond to temperature and act on their temperature preference. Moreover, similar principles of bidirectional sensing might be used by bacteria to locate optimal points in gradients of other physical or chemical stimuli.

## Materials and methods

### Strains and plasmids

All *Escherichia coli* K12 strains and plasmids used in this study are listed in *Table 1* and *Table 2*, respectively. Strains used for FRET analyses were derived from RP437 (*Parkinson and Houts, 1982*). VS223 (Δ*cheY-cheZ*, Δ*flgM*), in this study referred to as wildtype, was transformed with pVS88 (*cheZ-ecfp/cheYeyfp*) - coexpressing *cheZ-ecfp/cheYeyfp* from a bicistronic mRNA; VS181 (Δ(*cheY cheZ*) Δ*tsr* Δ(*tar tap*) Δ*trg* Δ*aer*) transformed with pVS88 and pVS121 (*tar*[EEEE]) or pVS362 (*tsr*[EEEE]) for receptor production and referred to as Tar-only and Tsr-only strain, respectively. Alternatively, for protein quantification VH1 (Δ(*cheR cheB cheY cheZ*) Δ*tsr* Δ(*tar tap*) Δ*trg* Δ*aer*) was transformed with pVS88 and receptors in different modification states (see *Table 1*). For behavioral analysis *E. coli* AW405 (HCB1) (*Armstrong et al., 1967*) was transformed with pCMW1 (*Girgis et al., 2007*) for *gfp* expression. Growth analysis was performed using *E. coli* MG1655.

### Growth conditions

All strains used for FRET and microfluidics were grown aerobically in tryptone broth (TB; 1% tryptone, 0.5% NaCl, pH 7.0) at 275 rpm at 34°C as described previously (*Neumann et al., 2010*). Briefly, cells diluted 1:100 in TB from an overnight culture, supplemented with appropriate antibiotics (100 µg/ml ampicillin, 17 µg/ml chloramphenicol or 50 µg/ml kanamycin) and inducers (isopropyl β-D-thiogalactoside (IPTG) or sodium salicylate; see *Table 2*) were grown to an optical density $OD_{600}$ of 0.6 for FRET and protein quantification or $OD_{600}$ of 0.35 for microfluidics, unless indicated otherwise. For analysis of growth at different temperatures, an overnight culture grown at the respective temperatures in M9 minimal medium (47 mM $Na_2HPO_4$, 22 mM $KH_2PO_4$, 8 mM $NaCl_2$, 2 mM $MgSO_4$, 100 µM $CaCl_2$, 1x trace metals (TEKnova, Hollister, CA) and 0.2% glycerol, pH7) was diluted 1:200 and incubated with shaking at 217 rpm in a total volume of 110 µl in a 96-well plate (Greiner Bio-One, Frickenhausen, Germany) and $OD_{600}$ was measured using a plate reader (M200 Absorbance, Tecan, Männedorf, Switzerland). Where indicated, M9 medium was supplemented with amino acids. Growth was analyzed calculating the growth delay (time from inoculation to the half-maximal $OD_{600}$) or the growth rate using a custom-written MATLAB (MathWorks, Natick, MA) script (see *Source code file 3*- Growth).

### FRET experiments

Cell preparation and FRET measurements were done as prescribed previously for chemotactic stimulation (*Neumann et al., 2010*; *Sourjik and Berg, 2002a*; *Sourjik et al., 2007*). Briefly, cells harvested at mid-exponential growth phase were washed with tethering buffer (10 mM $KPO_4$, 0.1 mM EDTA, 1 µM methionine, 10 mM lactic acid, pH 7) and stored at 4°C for 20 min to inhibit protein synthesis. Cells were 100-fold concentrated, attached to a polylysine-coated coverslip and placed in a flow

**Table 1.** Strains.

| Strains | Relevant genotype or phenotype | Reference |
|---------|-------------------------------|-----------|
| RP437 | *Escherichia coli* K12 derivative; wild type for chemotaxis | (*Parkinson and Houts, 1982*) |
| AW405 | *Escherichia coli* K12 derivative; wild type for chemotaxis | (*Armstrong et al., 1967*) |
| MG1655 | *Escherichia coli* K12 | (*Blattner et al., 1997*) |
| VH1 | Δ(*cheR cheB cheY cheZ*) Δ*tsr* Δ(*tar tap*) Δ*trg* Δ*aer* | (*Endres et al., 2008*) |
| VS181 | Δ(*cheY cheZ*) Δ*tsr* Δ(*tar tap*) Δ*trg* Δ*aer* | (*Sourjik and Berg, 2002b*) |
| VS168 | Δ*cheA* Δ(*cheY cheZ*) | (*Sourjik and Berg, 2002a*) |
| VS223 | Δ*cheY-cheZ*, Δ*flgM* | this work |

DOI: https://doi.org/10.7554/eLife.26607.045

**Table 2.** Plasmids used for FRET analyses.

| Plasmids | Relevant genotype | Resistance | Replication origin | Induction | Reference |
|---|---|---|---|---|---|
| pVS88 | cheZ-ecfp/cheYeyfp | ampicillin | pBR | 50 µM IPTG | (*Sourjik and Berg, 2004*) |
| pVS120 | tar [QEEE] | chloramphenicol | pACYC | 2 µM sodium salicylate | (*Sourjik and Berg, 2004*) |
| pVS121 | tar [EEEE] | chloramphenicol | pACYC | 1 µM sodium salicylate | (*Sourjik and Berg, 2004*) |
| pVS122 | tar [QEQQ] | chloramphenicol | pACYC | 2 µM sodium salicylate | (*Sourjik and Berg, 2004*) |
| pVS123 | tar [QEQE] | chloramphenicol | pACYC | 2 µM sodium salicylate | (*Sourjik and Berg, 2004*) |
| pVS415 | tar [QQQQ] | chloramphenicol | pACYC | 2 µM sodium salicylate | (*Endres et al., 2008*) |
| pVS 160 | tsr [QEQE] | chloramphenicol | pACYC | 2 µM sodium salicylate | (*Oleksiuk et al., 2011*) |
| pVS 356 | tsr [QEEE] | chloramphenicol | pACYC | 2 µM sodium salicylate | (*Oleksiuk et al., 2011*) |
| pVS 357 | tsr [QEQQ] | chloramphenicol | pACYC | 2 µM sodium salicylate | (*Oleksiuk et al., 2011*) |
| pVS361 | tsr [QQQE] | chloramphenicol | pACYC | 2 µM sodium salicylate | (*Oleksiuk et al., 2011*) |
| pVS362 | tsr [EEEE] | chloramphenicol | pACYC | 0.6 µM sodium salicylate | (*Oleksiuk et al., 2011*) |

DOI: https://doi.org/10.7554/eLife.26607.046

chamber under constant flow (500 µl/min) for the entire experiment. Temperature in the flow cell was controlled using a Peltier element (*Figure 1—figure supplement 1A*) (*Oleksiuk et al., 2011*). Cells were adapted in tethering buffer (with or without attractants) for at least 25 min at a constant flow and at 21°C before stimulation with indicated steps of temperature. The rate of temperature change could be readily monitored due to the direct and immediate effect of temperature on the fluorescence of CFP and YFP (*Figure 1—figure supplement 1E Inset*). Fluorescence of 300–500 cells was continuously recorded in the yellow (HQ535/30) and cyan (D485/40) channels using photon counters (Hamamatsu, Hamamatsu City, Japan) with 1.0 s integration time, using a custom-modified Zeiss Axio Imager.Z1 fluorescence microscope. FRET response was measured as the change in the ratio of yellow to cyan fluorescence due to energy transfer and normalized to the response of buffer-adapted cells to saturating stimulation with chemical attractant, either α-methyl-DL-aspartate (MeAsp; Sigma Aldrich, Taufkirchen, Germany) or L-serine (Acros Organics – Thermo Fisher Scienitifc, Nidderau, Germany), at 21°C.

## Microfluidics

Cell preparation was done using a filtration method as described previously by Berg and Turner at room temperature (*Berg and Turner, 1990*). Cells were washed in tethering buffer and within 30 min of cell harvesting a diluted cell suspension ($OD_{600}$ = 0.002) was loaded into the microfluidics device made of polydimethylsiloxane (PDMS). The device consists of a central assay channel with a width of 500 µm, flanked on either side by water circulation channels, with cold water circulated through the channel on the left side and the warm circulated through the channel on the right side (*Figure 1—figure supplement 1B*), resulting in a near-linear temperature gradient between these last two channels (*Figure 1—figure supplement 3*). Prior to loading with washed cells, the channel of the device was flushed with 0.1% BSA (BioLabs B9001S, New England BioLabs, Ipswitch, MA) and EtOH for one hour, followed by tethering buffer for 4 hr. For adaptation MeAsp and L-serine were added to the buffer as stated in the text. The profile of the thermal gradient was measured and calibrated as a function of the spectral shifts of 50 µM pH-sensitive 2′,7′ bis (two carboxyethyl) 5 (and 6) carboxyfluorescein (BCECF) and 50 mM Tris buffer (pH 7.1). To that end fluorescence intensities were measured at two excitation wavelengths, 490 nm, and 440 nm, with fixed emission at 535 nm. The ratio of these two fluorescent intensities was used for calibration.

## Cell tracking

Cells were imaged ~500 µm from the end of the assay channel and, 15 µm above the cover slip, at a flow rate of 30 µm/second. Time-lapsed images were taken at 12 frames/second for 30 min using an inverted microscope (Nikon Eclipse Ti-U, 20x objective, EGFP/mRFP-1 filter cube, EXFO X-Cite 120Q lamp) coupled with a CCD camera (Stingray F145B), and custom software written in LabVIEW

(National Instruments, Austin, TX). Under used settings, each pixel of the movie corresponds to 0.3 μm.

Data analysis was performed using custom scripts written in MATLAB (see *Source code file 2* – Microfluidics). Cells were identified using standard machines vision techniques, first by image thresholding for particle identification and then by filtering out particles less than five pixels, or with an eccentricity greater than 0.95. Trajectories of cells were compiled from their 2D centroid position using nearest neighbor criteria to concatenate these positions. To remove trajectories from non-motile cells, we filtered them based on their ratio of standard deviation in the *X* and *Y* directions, since non-motile cells moved very little in the *X* direction. Cells below the empirical threshold std(*Y*)/std(*X*) of 18 were defined as motile. Furthermore, all trajectories with a path length shorter than 10 pixels were eliminated from further analysis. Examples for typical trajectories are shown in *Figure 1— figure supplement 1B*. The *X*-positions from all viable trajectories were binned and normalized by the number of measurements to produce the thermotaxis histograms. This plot shows the relative frequency of occurrence of various *X*-positions of swimming cells at the downstream end of the microfluidic channel.

The thermotaxis migration coefficient (TMC) was used as a metric of cell tendency to migrate toward the warm (+1) or cold (−1) side of the microfluidics channel. TMC was defined as

$$TMC = -2 * [(mean(X) - X_{min})/(X_{max} - X_{min}) - 0.5]$$

where *X* is the array of all *X*-positions of viable paths, $X_{min}$ is left-side cutoff in the microfluidics channel (expressed in pixels), and $X_{max}$ is the right-side cutoff.

## Analysis of receptor modification

The extent of receptor methylation was determined using quantitative immune blotting as described in (*Neumann et al., 2010*). Cells were treated as described above for FRET and concentrated to OD$_{600}$ of approx. 13, adapted for 30 min with or without attractant at 21°C, and subsequently incubated for 20 min at indicated temperature. The reaction was stopped by addition of 4 × 95°C Laemmli buffer, samples were boiled and separated using 40 cm long 8% SDS-PAGE. Proteins were transferred to a 0.2 mm Hybond ECL nitrocellulose membrane by tank blotting and detected using an α-Tar antibody as primary antibody and a IRDyes 800-conjugated secondary antibody (Rockland, Limerick, PA) using an Odyssey Imager (LI-COR, Bad Homburg, Germany). Protein bands were quantified using the line-scan tool in ImageJ (http://rsbweb.nih.gov/ij).

## Mathematical modeling

In the free-energy model for receptor activity, the probability $A$ that a team of receptors will be active is $A = 1/(1 + e^F)$, where $F = \sum_i f_i$ is the sum of free-energy differences between active and inactive states of each of the receptors participating in the team, expressed in units of the thermal energy $k_B T$. Following Jiang et al. (*Jiang et al., 2009*) and Oleksiuk et al. (*Oleksiuk et al., 2011*), the free-energy difference for a receptor is given by

$$f_i = f_0([L]) + (T - T_0)f_1 - [g_0 + (T - T_0)g_1]m, \tag{1}$$

where $f_i$ has been expanded to first order in the temperature difference from a reference temperature $T_0$ (with the index $i$ suppressed on the RHS) (*Jiang et al., 2009*; *Oleksiuk et al., 2011*). Here, $f_0([L]) = f_0^{(0)} + \log\left[\left(1 + \frac{[L]}{K_{off}}\right)/\left(1 + \frac{[L]}{K_{on}}\right)\right]$, where $f_0^{(0)}$ is the free-energy difference in the absence of ligand, [L] is the ligand concentration specific to the receptor type, and $K_{on/off}$ are the receptor-ligand dissociation constants in the active (*on*) and inactive (*off*) states. For a chemoattractant, $K_{off} < K_{on}$. For simplicity, the free-energy difference is assumed to depend linearly on the methylation level of the receptor dimer, $m = 0, \ldots m_{tot} (= 8)$, with a coefficient $g = [g_0 + (T - T_0)g_1] > 0$ (*Endres et al., 2007*; *Shimizu et al., 2010*). In principle, all these parameters could be different between the Tar and Tsr receptors, but here we take them to be identical, except for the fact that the free energy of the Tar receptor is only affected by the presence of MeAsp, while that of Tsr is affected only by serine. For simplicity, we assumed the basic allosteric signaling unit to be a trimer of receptors with randomly mixed Tar and Tsr (*Ames et al., 2002*; *Hansen et al., 2010*).

In such a trimer, each of the three receptors can be either Tsr or Tar with probability reflecting their relative expression levels (1:1 in our case).

Following the model of Meir et al. (**Meir et al., 2010**) for slowing down of methylation due to the scarcity of methylation sites, the kinetics of methylation is determined by

$$\frac{dm(t)}{dt} = \gamma R \frac{m_{\text{tot}} - m(t)}{m_{\text{tot}} - m(t) + N_0} [1 - A(t)] - \gamma B \frac{m(t)}{m(t) + N_0} A(t),$$

(2)

where $N_0$ is the parameter that determines how abruptly methylation and demethylation slow down as the receptors approach the saturated limits $m = 8$ and $m = 0$, respectively. The adapted steady-state methylation level $m_{ss}$ obeys

$$\frac{m_{\text{tot}} - m_{ss}}{m_{\text{tot}} - m_{ss} + N_0} \frac{1 - A}{1 - A_0} = \frac{m_{ss}}{m_{ss} + N_0} \frac{A}{A_0}$$

(3)

where $A_0$ is the adapted activity in the absence of slowing down of methylation/demethylation ($N_0 = 0$, $0 \leq m \leq 8$). Here we assume that all receptor dimers in the trimer have the same adapted methylation level $m$, which is obtained by solving **Equation (3)** for a given ligand concentration and a given temperature, in view of the above dependence of the probability $A$ on $m$, $[L]$, and $T$. The average activity $\bar{A}$ is given by averaging this probability over all trimers. For the case of mixed receptors, we assume equal fractions of Tar and Tsr with trimer probabilities reflecting random mixing. Unless indicated otherwise, the parameters used for simulations were $A_0$=1/3 (a typical wild-type adapted activity; (**Neumann et al., 2012**; **Sourjik and Berg, 2002b**; **Sourjik et al., 2007**), $N_0$=2 (which corresponds to the saturation parameter previously obtained in Meir et al. (**Meir et al., 2010**) for $T$ = 34°C in the current model), $T_0$ = 24°C (which is a convenient arbitrary reference temperature), $f_1$=1.2, $g_0$=0, and $g_1$=0.2 (where these $f$ and $g$ parameters are chosen to yield an accumulation temperature near the range observed experimentally). For additional model details see Supplementary material. The thermotactic response at temperature $T$ is given by $[\bar{A}(T+3) - \bar{A}(T)]/\Delta\bar{A}$, where the response, as in the experiment, is normalized by the change of activity $\Delta A$ at the lowest temperature upon saturation by ligand.

## Supplementary material
### Mathematical modeling

According to **Equation 1**, the change in the free-energy difference of a single receptor in response to a temperature change is proportional to $f_1 - mg_1$, where $m$ is the methylation level. For a team of identical heat-seeking receptors, as temperature increases, approaches from below its asymptotic high-temperature value, $f_1/g_1$, and the response to a temperature change vanishes. Similarly, for a team of identical cold-seeking receptors, as temperature increases $m$ approaches this same asymptotic value from above. This implies that there is no accumulation temperature for teams of identical receptors, but rather a gradual decrease of thermal response with increasing temperature, with methylation $m$ approaching $f_1/g_1$. For example, **Figure 4 – Figure Supplement 1A** depicts the modeled average per-receptor methylation level of Tar-only teams, whose modeled activity is depicted in **Figure 4A**.

As can be seen in **Figure 4 – Figure Supplement 1A**, at a given temperature the adapted methylation increases with ligand concentration, while as temperature increases the adapted methylation approaches its asymptotic value. Note that the highly methylated receptors adapt more weakly in response to temperature changes, because of the invoked slowing down of methylation near saturation.

When there is more than one receptor type in the cell, then receptor trimers consist of different combinations receptors, for example, of Tar and Tsr. In the absence of slowing down of methylation, the above argument still holds, implying the absence of an accumulation temperature. This means that even though different trimers may have different temperature responses, their adaptive methylations compensate each other, and the thermal response retains the same sign with increasing temperature (**Figure 4 – Figure Supplement 2**).

On the other hand, if one invokes slowing down of methylation near saturation, then the trimers that are close to saturation cannot fully compensate for the adaptive methylation of the trimers whose methylation levels are far from saturation, and the thermal response may change sign.

*Figure 4 – Figure Supplement 1B* depicts the modeled methylation levels of the different trimers, for conditions corresponding to the dark green thermal-response curve in *Figure 4B*.

As can be seen in *Figure 4 – Figure Supplement 1B*, the Tsr-only trimer is not affected by the large amount of MeAsp, and its methylation level remains very low, while the methylation levels of the Tar-containing trimers are close to saturation due to failure of precise adaptation. At low temperatures, the thermal response is dominated by the Tsr-only trimers, and the cell is heat seeking. However, as temperature increases, the methylation of the Tsr-only trimer approaches the asymptotic methylation and its temperature response decreases. On the other hand, because of the slowing down of methylation near saturation, the methylation levels of the Tar-containing trimers hardly change, and consequently their temperature response remains practically constant, thus eventually dominating the response, which leads to a change of sign of the thermal response, with the cell becoming cold seeking. This mechanism of response inversion is schematically illustrated in *Figure 4 – Figure Supplement 1C*.

## Analysis of amino acid composition

*E. coli* cells were grown in M9CG media (M9 media supplemented with casamino acids, 1 g/L) and glucose (4 g/L)) as described in Yoney and Salman (*Yoney and Salman, 2015*). Briefly, cells from an overnight culture (30°C in M9CG) were diluted 1:50 in 10 mL M9CG, and the culture was grown to OD of 0.6 at 30°C. Cells were then harvested by centrifugation, resuspended in fresh M9CG at OD of 0.3 and incubated for 40 min. To extract metabolites cultures were vacuum-filtered through a 0.45 μm pore size filter and 100 μL of extract was transferred into 400 μL 50:50 acetonitrile/methanol cooled to −20°C and centrifuged at 13,000 rpm for 10 min and the supernatant was directly used for LC-MS/MS. Data were scored using the correlation between the $^{12}$C and the $^{13}$C channel and normalized to internal standards of amino acids as described previously (*Guder et al., 2017*).

## Acknowledgements

The authors thank Sean Murray for help with data analysis of the growth experiments, Remy Colin and Sean Murray for critically reading the manuscript and Hannes Link for the LC-MS/MS analysis of the amino acid composition. This work was supported in parts by grant R01 GM082938 from the National Institutes of Health, by grant 294761-MicRobE from the European Research Council and by grant PHY-1411313 from the National Science Foundation.

## Additional information

### Funding

| Funder | Grant reference number | Author |
|---|---|---|
| National Institutes of Health | R01 GM082938 | Vladimir Jakovljevic<br>Yigal Meir<br>William S Ryu<br>Ned S Wingreen<br>Victor Sourjik |
| H2020 European Research Council | 294761-MicRobE | Vladimir Jakovljevic<br>Victor Sourjik |
| Max-Planck-Institut für Terrestrische Mikrobiologie | Open-access funding | Victor Sourjik |
| National Science Foundation | PHY-1411313 | Alex Groisman |

The funders had no role in study design, data collection and interpretation, or the decision to submit the work for publication.

### Author contributions

Anja Paulick, Conceptualization, Resources, Data curation, Formal analysis, Investigation, Visualization, Methodology, Writing—original draft, Writing—review and editing; Vladimir Jakovljevic, Conceptualization, Resources, Investigation, Methodology; SiMing Zhang, Investigation, Methodology;

Michael Erickstad, Resources, Methodology; Alex Groisman, Resources, Supervision, Methodology; Yigal Meir, Conceptualization, Resources, Data curation, Formal analysis, Funding acquisition, Investigation, Methodology, Writing—original draft, Writing—review and editing; William S Ryu, Conceptualization, Data curation, Formal analysis, Supervision, Funding acquisition, Investigation, Methodology, Writing—review and editing; Ned S Wingreen, Conceptualization, Supervision, Funding acquisition, Investigation, Methodology, Writing—original draft, Project administration, Writing—review and editing; Victor Sourjik, Conceptualization, Data curation, Supervision, Funding acquisition, Methodology, Writing—original draft, Project administration, Writing—review and editing

### Author ORCIDs
Anja Paulick ⓘD http://orcid.org/0000-0001-7103-6287
William S Ryu ⓘD http://orcid.org/0000-0002-0350-7507
Ned S Wingreen ⓘD http://orcid.org/0000-0001-7384-2821
Victor Sourjik ⓘD http://orcid.org/0000-0003-1053-9192

### Decision letter and Author response
Decision letter https://doi.org/10.7554/eLife.26607.051
Author response https://doi.org/10.7554/eLife.26607.052

## Additional files

### Supplementary files
• Source code file 1. Modelling
DOI: https://doi.org/10.7554/eLife.26607.047

• Source code file 2. Microfluidics
DOI: https://doi.org/10.7554/eLife.26607.048

• Source code file 3. Growth
DOI: https://doi.org/10.7554/eLife.26607.049

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
