## [Decision Letter]

Thank you for submitting your article "Mechanism of bidirectional thermotaxis in *Escherichia coli*" for consideration by *eLife*. Your article has been reviewed by two peer reviewers, and the evaluation has been overseen by a Reviewing Editor and Gisela Storz as the Senior Editor. The reviewers have opted to remain anonymous.

The reviewers have discussed the reviews with one another and the Reviewing Editor has drafted this decision to help you prepare a revised submission.

The manuscript "Mechanism of bidirectional thermotaxis in *Escherichia coli*" by Paulick et al. describes a study of the response of bacteria to temperature changes, and how this response can lead to the accumulation of the bacteria at a specific temperature in a gradient. The study combines FRET measurements of the CheA activity, measurements of cell movement in microfluidic channels across which a temperature gradient is applied, and a mathematical model that describes how the response of the signal transduction pathway to temperature changes is influenced by the presence of attractants and by the methylation state of the receptors. The manuscript is well-written and the results are for the most part clear and well-explained. The topic of the manuscript is very important and timely. There has been a lot of discussion of how bacteria sense and respond to temperature recently, but no clear answer has been provided yet, and this study advances this question significantly.

Several concerns should however be addressed before the manuscript is published:

1) The links between the findings and previous literature should be clarified in several instances:

A) One of the main findings of the study is that with the two ligands (Serine and MeAsp) present, the bacteria will not accumulate at a specific temperature, and only when one or the other is present then the bacteria would accumulate at a specific temperature. However, other studies (such as Yoney and Salman 2015) have claimed that in complex medium which contains the two ligands, the bacteria have a favored temperature at which they accumulate in a gradient. How do these results align with the previous results? Can the ratio of the ligands used in the experiments be important? I think that the authors should address this contradiction and explain if this is a specific case, or if it is a contradiction that should be clarified in future studies and with further tests?

B) Another contradiction between the results presented in this study and previous results is that Tar does not switch response from thermophilic to cryophilic. This has been observed previously in Mizuno and Imae 1984, albeit at the level of cell behavior and not CheA activity. I think that the authors should address this contradiction better. Do they think that there is another layer of control between CheY-P and the motor switching? Do they think that the dynamics of CheY-P binding to CheZ is different than its biding unbinding dynamics to the motor? Or do they think that the previous results are not accurate enough?)

C) One more contradiction between these results and previous results is that here, the authors do not observe switching in the response of wt bacteria in buffer, whereas in a previous publication of one of the co-authors (Paster and Ryu 2008) it was reported that the response (which again was measured at the motor level) switched direction around 37 degrees.

D) The authors claim here that the accumulation temperature of the bacteria is actually chosen to optimize their growth. This is based on the fact that addition of serine to the growth medium seems to have the least effect on the growth around the accumulation temperature of the bacteria (~30degrees). However, they also show that the accumulation temperature of the bacteria can change as a function of the O.D. (Figure 3—figure supplement 2) due to the change in the expression level of the receptors Tar and Tsr. How does this result integrate with the previous one – a test at the motor level?

2) Definitions, experimental methodology and analysis should be better explained:

A) The narrative has been built around the concept of an accumulation temperature. This term however, has not been explained in the text. It seems that the peaks in cell distributions (motility assays, Figure 2) are likely what the authors term as cell accumulation. The FRET data on the other hand provides information about cross-over temperatures, where the network's thermophilic response inverts to cryophilic. Brownian motion, drift, cell-filament characteristics, hydrodynamic interactions together ensure that the accumulation temperatures and the cross-over temperatures are related, but not quite the same. Hence, the repeated conflation of 'accumulation' with the FRET data throughout the text is questionable.

B) The variable thermal gradients and channel widths employed in the motility assays make it difficult to ascertain whether the peak cell density appears due to the receptor-level interactions as suggested, or whether it is a function of wide channels and steep thermal gradients. For example, the gradients in Figure 2 and Figure 2—figure supplement 2 are ~ 1.5 deg/100 μm to 2.5 degree/100 um. Compared to these, figures where peaks are missing employ about half the thermal gradient. Can the authors confirm whether the accumulation is not a function of steep gradients alone?

C) In general what are the dimensions of the microfluidic channel used for tracking cells under a gradient? Channels used in Figure 1—figure supplement 1 and Figure 2 seem to be different.

D) The experiments in the temperature gradient are not well explained. Maybe adding a figure showing typical trajectories of cells that were used to calculate the TMC, and showing the x and y directions in the channel can help. Also, explain how the normalized cell count in Figure 2 was obtained.

E) The driving force for the thermotactic response in the presence of the two attractants reduces at higher temperatures (Figure 2). It seems quite possible that under a treatment of 10:1 MeAsp/serine, the cells might avoid lower temperature regions and concentrate near higher temperatures (40-45 degrees). Would the authors consider that an accumulation? In other words, how sharp does the peak need to be in order to be termed as accumulation?

F) The temperature dynamics in the FRET experiments should be more clearly explained in a graph showing how fast the temperature increased, and how long it took to stabilize.

G) It is suggested in the Abstract and elsewhere that the model explains accumulation temperature, but quantitative predictions of either accumulation or cross-over temperatures are not presented. The model makes qualitative predictions that appear similar to the observed cross-over behavior but it quits unexpectedly at temperatures above 35 °C. Predictions over the entire temperature range employed in the experiments. How were the parametric values (–subsection “Mathematical modeling”, last paragraph) taken from previous works, fitting experiments? In general, the authors would greatly help the reader understand the model better by explaining the physics underlying the modulation of equilibrium methylation levels by changes in demethylation/methylation kinetics.

*Reviewer #2:*

The manuscript "Mechanism of bidirectional thermotaxis in *Escherichia coli*" by Paulick et al. describes a study of the response of bacteria to temperature changes, and how this response can lead to the accumulation of the bacteria at a specific temperature in a gradient. The study combines FRET measurements of the CheA activity, measurements of cell movement in microfluidic channels across which a temperature gradient is applied, and a mathematical model that describes how the response of the signal transduction pathway to temperature changes is influenced by the presence of attractants and by the methylation state of the receptors. The manuscript is well-written and the results are for the most part clear and well-explained. The topic of the manuscript is very important and timely. There has been a lot of discussion of how bacteria sense and respond to temperature recently, but no clear answer has been provided yet, and this study advances this question significantly.

I have however few questions that I think the authors should address before publication:

1) Some experimental details are lacking.a) The dimensions of the microfluidic channel used for tracking cells under a gradient is not clearly specified. In Figure 1—figure supplement 1 it looks like it is 500µm wide, which matches most graphs presented, however, in Figure 2 it seems to be different.

b) The experiments in the temperature gradient are not well explained. Maybe adding a figure showing typical trajectories of cells that were used to calculate the TMC, and showing the x and y directions in the channel can help. Also, explain how the normalized cell count in Figure 2 was obtained.

c) The temperature dynamics in the FRET experiments should be presented in a figure. How fast was the temperature increase, and how long until it stabilized? These details should be included.

2) The values used in the mathematical model should be explained. Why did the authors use these values in the simulations (subsection “Mathematical modeling”, last paragraph –)? Were they taken from a previous work? Were they obtained from fitting to experiments? Etc.

3) One of the main findings of the study is that with the two ligands (Serine and MeAsp) present, the bacteria will not accumulate at a specific temperature, and only when one or the other is present then the bacteria would accumulate at a specific temperature. However, other studies (such as Yoney and Salman 2015) have claimed that in complex medium which contains the two ligands, the bacteria have a favored temperature at which they accumulate in a gradient. How do these results align with the previous results? Can the ratio of the ligands used in the experiments be important? I think that the authors should address this contradiction and explain if this is a specific case, or if it is a contradiction that should be clarified in future studies and with further tests?

4) Another contradiction between the results presented in this study and previous results is that Tar does not switch response from thermophilic to cryophilic. This has been observed previously in Mizuno and Imae 1984, albeit at the level of cell behavior and not CheA activity. I think that the authors should address this contradiction better. Do they think that there is another layer of control between CheY-P and the motor switching? Do they think that the dynamics of CheY-P binding to CheZ is different than its biding unbinding dynamics to the motor? Or do they think that the previous results are not accurate enough?

5) One more contradiction between these results and previous results is that here the authors do not observe switching in the response of wt bacteria in buffer, whereas in a previous publication of one of the co-authors (Paster and Ryu 2008) it was reported that the response (which again was measured at the motor level) switched direction around 37 degrees. The questions raised in (4) also apply here, and I think that the authors should address this contradiction more clearly, and maybe add a test at the motor level.

6) The authors claim here that the accumulation temperature of the bacteria is actually chosen to optimize their growth. This is based on the fact that addition of serine to the growth medium seems to have the least effect on the growth around the accumulation temperature of the bacteria (~30degrees). However, they also show that the accumulation temperature of the bacteria can change as a function of the O.D. (Figure 3—figure supplement 2) due to the change in the expression level of the receptors Tar and Tsr. How does this result integrate with the previous one?

In general, I think that this is an excellent study and very important, which provides significant advancement to the field, but I think the authors should discuss some of these contradictions more, even if they cannot provide clear cut answers to these contradictions at this point.

*Reviewer #3:*

Paulick et al. have investigated an interesting response of the chemotaxis machinery to a non-chemical stimulus. The team's expertise in the FRET technique is clearly evident once again – the experimental data are solid. The conclusions, for most parts, are well-founded. The observed inversion of the chemotaxis machinery from a thermophilic to a cryophilic response has been modeled by taking into account an interaction between the tar and tsr receptors. This work addresses several puzzling aspects of prior studies and is likely to be of wide interest.

1) The narrative has been built around the concept of an accumulation temperature. This term however, has not been explained in the text. It seems that the peaks in cell distributions (motility assays, Figure 2) are likely what the authors term as cell accumulation. The FRET data on the other hand provides information about cross-over temperatures, where the network's thermophilic response inverts to cryophilic. Brownian motion, drift, cell-filament characteristics, hydrodynamic interactions together ensure that the accumulation temperatures and the cross-over temperatures are related, but not quite the same. Hence, I don't agree with the repeated conflation of 'accumulation' with the FRET data throughout the text.

2) The variable thermal gradients and channel widths employed in the motility assays make it difficult to ascertain whether the peak cell density appears due to the receptor-level interactions as suggested, or whether it is a function of wide channels and steep thermal gradients. For example, the gradients in Figure 2 and Figure 2—figure supplement 2 are ~ 1.5 deg/100 μm to 2.5 degree/100 um. Compared to these, figures where peaks are missing employ about half the thermal gradient. Can the authors confirm whether the accumulation is not a function of steep gradients alone?

3) The driving force for the thermotactic response in the presence of the two attractants reduces at higher temperatures (Figure 2). It seems quite possible that under a treatment of 10:1 MeAsp/serine, the cells might avoid lower temperature regions and concentrate near higher temperatures (40-45 degrees). Would the authors consider that an accumulation? In other words, how sharp does the peak need to be in order to be termed as accumulation?

4) It is suggested in the Abstract and elsewhere that the model explains accumulation temperature, but I did not find any quantitative predictions of either accumulation or cross-over temperatures. The model makes qualitative predictions that appear similar to the observed cross-over behavior but I can't tell for sure since it quits unexpectedly at temperatures above 35 °C. I request an inclusion of predictions over the entire temperature range employed in the experiments and a note on how the parametric values (subsection “subsection “Mathematical modeling”, last paragraph) were determined. Most importantly, the authors would greatly help the reader understand the model better by explaining the physics underlying the modulation of equilibrium methylation levels by changes in demethylation/methylation kinetics.

---

## [Author Response]

*[…] Several concerns should however be addressed before the manuscript is published:*

*1) The links between the findings and previous literature should be clarified in several instances:*

*A) One of the main findings of the study is that with the two ligands (Serine and MeAsp) present, the bacteria will not accumulate at a specific temperature, and only when one or the other is present then the bacteria would accumulate at a specific temperature. However, other studies (such as Yoney and Salman 2015) have claimed that in complex medium which contains the two ligands, the bacteria have a favored temperature at which they accumulate in a gradient. How do these results align with the previous results? Can the ratio of the ligands used in the experiments be important? I think that the authors should address this contradiction and explain if this is a specific case, or if it is a contradiction that should be clarified in future studies and with further tests?*

We believe that the discrepancy between our study and the Yoney and Salman paper is only apparent, and resides in the interpretation of the results. As rightly pointed out by the reviewer, the differences in the observed behavior are most likely explained by different relative levels (as well as signaling strength) of ligands used by the Salman group.

In the experiments done by the Salman group, *E. coli* behavior in temperature gradients was studied upon prolonged (>30 min) incubation in complex medium that contains a mixture of casamino acids, which should contain submillimolar levels of aspartate and serine. Although such combination of submillimolar levels of aspartate and serine would indeed be expected to fully invert the response to cryophilic based on our results, several previous studies have shown that *E. coli* rapidly consumes serine from the culture medium. Given the reported rate of serine consumption, only (sub-) μM amounts of serine should be present in the medium after 30 min incubation with *E. coli* culture. Thus, bacteria in these experiments are in fact primarily stimulated with Tar ligand aspartate and only very weakly with the Tsr ligand serine, and the accumulation temperature produced by such asymmetric stimulation is consistent with our data. To verify this hypothesis, we have now analysed the amino acid (serine) composition by LCMS/MS as described in Guder et al., 2016. Cells were grown to an OD 0.6 (diluted to OD 0.3) and incubated for 40min in M9CG media used in the studies by Yoney et al., 2015. We found that serine was taken up quickly, and its concentration decreased more than 10-fold, from ~22 µM to ~2µM, after the 40 min incubation time (new Figure 3—figure supplement 3). Similar results were reported before by Yang et al., 2015.

Yoney and Salman have further shown that addition of 1 mM glycine, another ligand of Tsr, to this culture does not abolish accumulation. Furthermore, in the *Δtar* background (where Tsr dominates the response), they observed reduction but not inversion of the thermophilic response upon addition of increasing concentrations of glycine. While this appears to contradict our results showing that the response of Tsr to temperature can be inverted similarly to that of Tar, and costimulation with high levels of Tar and Tsr ligands should invert the response to cryophilic over the entire temperature range, this apparent contradiction can be easily resolved by taking into account that glycine is a much (~1000-fold) weaker attractant than serine (Yang et al., 2015). Consequently, stimulation with sub-millimolar and low millimolar levels of glycine used in the Yoney and Salman study will not increase the methylation of Tsr to the levels required for the cryophilic response. We now directly confirmed this conclusion by measuring the effects of glycine on the temperature response in FRET experiments. We found that at glycine concentrations of up to 1 mM, the temperature response of the Tsr-only strain is either thermophilic or non-detectable. The response of the Tsr-only strain became weakly cryophilic only when the concentration of glycine was increased to the very high level of 30 mM (new Figure 3—figure supplement 2).

We thank the reviewer for pointing out the need to address these discrepancies in more detail, and we have now included this discussion in the Results and Discussion sections and added two new supplementary Figures (Figure 3—figure supplement 2 and Figure 3—figure supplement 3).

B) Another contradiction between the results presented in this study and previous results is that Tar does not switch response from thermophilic to cryophilic. This has been observed previously in Mizuno and Imae 1984, albeit at the level of cell behavior and not CheA activity. I think that the authors should address this contradiction better. Do they think that there is another layer of control between CheY-P and the motor switching? Do they think that the dynamics of CheY-P binding to CheZ is different than its biding unbinding dynamics to the motor? Or do they think that the previous results are not accurate enough?)

Indeed, previous studies, including the paper by Mizuno and Imae, suggested that the dependences of the thermal responses of Tsr and Tar on their levels of methylation are different. They concluded that Tar functions as a warm sensor in low methylation states but as a cold sensor in high methylation states, whereas Tsr was suggested to similarly function as a warm sensor in low methylation states but to lose its temperature sensitivity in high methylation states. A similar conclusion has been drawn by Yoney and Salman upon stimulating a *Δtar* strain with glycine. While this latter result can be explained when taking into account that glycine is a much weaker attractant (see above), the results of Mizuno and Imae are indeed more difficult to reconcile with our work. We do not think that there is a major effect of the control layer between CheY phosphorylation and motor switching on the overall temperature response, since the response of the wild-type cells observed in the FRET assays is consistent with the response of swimming cells in microfluidic devices. In our opinion, the most likely explanation is, indeed, simply insufficient accuracy of the response measurements done by Mizuno and Imae, where only ~100 tracks were manually scored per data point. Moreover, whereas for Tar the thermophilic and cryophilic responses observed in our FRET measurements have similar magnitudes (Figure 3), the cryophilic response observed for Tsr is weaker than the thermophilic response (Figure 3), making the cryophilic response more difficult to detect. This relative weakness of the cryophilic response might explain why Mizuno and Imae could detect the Tar-mediated cryophilic response but missed the Tsr-mediated cryophilic response. The effect of serine in the experiments of Mizuno and Imae might have been further weakened due to its consumption (see Editor’s point 1A; although the extent of consumption is difficult to assess because the experimental description in their paper does not specify how long the cell suspension was incubated in the presence of serine) and by higher optical density of the culture used in their experiments (which lowers the level of Tsr expression, see Editor’s point 1D).

We mentioned this discrepancy in the initial version of the manuscript (both in the Results and in the Discussion sections), and in the revised manuscript, we have now expanded the discussion of its possible explanations.

*C) One more contradiction between these results and previous results is that here, the authors do not observe switching in the response of wt bacteria in buffer, whereas in a previous publication of one of the co-authors (Paster and Ryu 2008) it was reported that the response (which again was measured at the motor level) switched direction around 37 degrees.*

As mentioned in the Introduction, the major difference between the experiments performed in our current study and in the previous studies of the thermotactic response, including that by Paster and Ryu 2008, is that both the FRET and microfluidic experiments were performed under continuous flow to avoid secondary effects, such as those due to the accumulation of secreted attractants or to the consumption of oxygen. As described in the Materials and methods of Paster and Ryu, tethered cells used in their experiments were highly concentrated on the tunnel slide and incubated for up to 3.5 hours. It is thus very possible that switching of the thermal response observed by Paster and Ryu was caused by such residual chemotactic stimulation. This point is now stated more explicitly in the Discussion.

*D) The authors claim here that the accumulation temperature of the bacteria is actually chosen to optimize their growth. This is based on the fact that addition of serine to the growth medium seems to have the least effect on the growth around the accumulation temperature of the bacteria (~30degrees). However, they also show that the accumulation temperature of the bacteria can change as a function of the O.D. (Figure 3—figure supplement 2) due to the change in the expression level of the receptors Tar and Tsr. How does this result integrate with the previous one – a test at the motor level?*

We agree that this point merits more discussion and thank the reviewer for raising it. We indeed show that the density of the bacterial culture affects the response inversion in the presence of serine, such that as the culture OD is increased, the switching of the response increases in temperature, eventually, at 100 µM serine, the response remains thermophilic over the entire temperature range (Figure 3—figure supplement 4 in the revised version of the manuscript). As a matter of fact, we believe that such disappearance of the serine-induced accumulation temperature at high OD might be physiologically meaningful. As mentioned above, serine is readily consumed from the medium (as the first amino acid), and consequently, *E. coli* cultures with high OD should not be strongly affected by serine toxicity. It is further possible that the shift of the accumulation temperature in the intermediate range of OD might be mirrored by a gradual change in the optimal growth temperature at lower serine concentrations. However, due to substantial day-to-day variability of the growth curves, our experiments cannot provide sufficient resolution to test this hypothesis.

*2) Definitions, experimental methodology and analysis should be better explained:*

*A) The narrative has been built around the concept of an accumulation temperature. This term however, has not been explained in the text. It seems that the peaks in cell distributions (motility assays, Figure 2) are likely what the authors term as cell accumulation. The FRET data on the other hand provides information about cross-over temperatures, where the network's thermophilic response inverts to cryophilic. Brownian motion, drift, cell-filament characteristics, hydrodynamic interactions together ensure that the accumulation temperatures and the cross-over temperatures are related, but not quite the same. Hence, the repeated conflation of 'accumulation' with the FRET data throughout the text is questionable.*

We thank the reviewer for pointing out that the term ‘accumulation temperature’ and the possible difference between the accumulation temperature and the cross-over temperature observed in FRET experiments were not sufficiently well explained in the text. In general, the agreement between FRET and microfluidics experiments in our study strongly suggests that direct effects of temperature on cell motility play only a minor role in *E. coli* thermotaxis. Nevertheless, we fully agree with the reviewer that these effects might slightly shift the accumulation temperature relative to the cross-over temperature, which might explain the result of our microfluidic experiment where cell accumulation in the presence of serine (Figure 2) seems to occur at a slightly lower temperature than the cross-over in FRET (Figure 2). We have modified the text accordingly to clarify these points.

*B) The variable thermal gradients and channel widths employed in the motility assays make it difficult to ascertain whether the peak cell density appears due to the receptor-level interactions as suggested, or whether it is a function of wide channels and steep thermal gradients. For example, the gradients in Figure 2 and Figure 2—figure supplement 2 are ~ 1.5 deg/100 μm to 2.5 degree/100 um. Compared to these, figures where peaks are missing employ about half the thermal gradient. Can the authors confirm whether the accumulation is not a function of steep gradients alone?*

We realized that there was a mistake in labeling Figure 2, as the channel width was in fact the same in all experiment. We sincerely apologize for this mistake. Nevertheless, the reviewer is absolutely correct in pointing out that the gradients in Figure 2 and Figure 2—figure supplement 2 are steeper than those used in Figure 1 and Figure 2. Such steeper gradients were used on purpose, to make it easier to identify the accumulation peak by expanding the range of temperatures to include regions of thermophilic and cryophilic response and thus increasing the ratio between the bacterial counts at the peak and away from it. Note that the width of the peak in Figure 2 is similar to the entire temperature range in Figure 2. Thus, the peak would be much less visible in the shallow temperature gradient. We have now mentioned this fact explicitly in the text. To further back up our claim that the observed accumulation is not simply a consequence of the greater gradient steepness, we have further expanded Figure 2—figure supplement 2 to show experiments performed in the same temperature gradient but either in the buffer (A) or in 1 µM (B) or 10 µM serine (C). The control measurement in the buffer shows no accumulation, despite the steeper gradient. We thank the reviewer for bringing this issue to our attention.

Additionally, to make the results of the FRET and microfluidic measurements easier to compare, we modified all figures related to microfluidics such that the warmer side of the channel is always the right side.

*C) In general what are the dimensions of the microfluidic channel used for tracking cells under a gradient? Channels used in Figure 1—figure supplement 1 and Figure 2 seem to be different.*

We thank the reviewers for pointing out this mistake in the labeling of Figure 2. As mentioned above, all channels have the same size, with 500 µm width, as illustrated in Figure 1—figure supplement 1. This mistake in labeling of Figure 2 is now corrected, and we apologize for it.

*D) The experiments in the temperature gradient are not well explained. Maybe adding a figure showing typical trajectories of cells that were used to calculate the TMC, and showing the x and y directions in the channel can help. Also, explain how the normalized cell count in Figure 2 was obtained.*

Cell tracking in the temperature gradient is described in detail in the Materials and methods section. Nevertheless, as suggested by the reviewer we have now added a panel showing typical trajectories to Figure 1—figure supplement 1. We also now describe the normalization of the data in the Materials and methods section – we apologize that it was missing in the initial version of the manuscript.

*E) The driving force for the thermotactic response in the presence of the two attractants reduces at higher temperatures (Figure 2). It seems quite possible that under a treatment of 10:1 MeAsp/serine, the cells might avoid lower temperature regions and concentrate near higher temperatures (40-45 degrees). Would the authors consider that an accumulation? In other words, how sharp does the peak need to be in order to be termed as accumulation?*

We have now clarified the definition of accumulation temperature in the text (see Editor’s point 2A), to emphasize that it refers to bidirectional accumulation towards a specific temperature. This is not the case in the example mentioned by the reviewer, and we now specifically state so in the text. We thank the reviewer again for pointing out that our definition of accumulation temperature was not sufficiently clear.

F) The temperature dynamics in the FRET experiments should be more clearly explained in a graph showing how fast the temperature increased, and how long it took to stabilize.

The temperature dynamics can be seen in Figure 1—figure supplement 1, which shows the *ΔcheA* strain. Because this strain has no specific chemotactic response to temperature, the only effect of temperature for this strain is the change in fluorescence of YFP and CFP. Since the effect of temperature on the efficiency of fluorescence (absorption coefficient and quantum yield) of a fluorophore is nearly immediate, the dynamics of variation of the fluorescence levels (or their ratio) provides a direct readout for the dynamics of temperature in our experiments. We now mention this point more explicitly in the text and have added an inset to Figure 1—figure supplement 1 showing this change for one temperature step at higher resolution. The onset time of the temperature changes is ~100 ms.

*G) It is suggested in the Abstract and elsewhere that the model explains accumulation temperature, but quantitative predictions of either accumulation or cross-over temperatures are not presented. The model makes qualitative predictions that appear similar to the observed cross-over behavior but it quits unexpectedly at temperatures above 35 °C. Predictions over the entire temperature range employed in the experiments. How were the parametric values (–subsection “Mathematical modeling”, last paragraph) taken from previous works, fitting experiments? In general, the authors would greatly help the reader understand the model better by explaining the physics underlying the modulation of equilibrium methylation levels by changes in demethylation/methylation kinetics.*

To clarify the model and its parameters, we have made several edits to the text, including the Abstract, modeling section of the Results, and modeling section of the Materials and methods. The basic mechanism leading to accumulation temperature is now illustrated in a new schematic figure (Figure 4—figure supplement 1). We have also more clearly stated in the text that the purpose of the model is to provide insight into the mechanism leading to accumulation temperature, not to quantitatively account for the experimental data. Nevertheless, we have expanded the temperature range used in simulations to 42 °C, as suggested by the reviewer, which does not affect our conclusions.